# A bidirectional network for appetite control in larval zebrafish

Caroline Lei Wee[1,2†‡]*, Erin Yue Song[1†], Robert Evan Johnson[1,2], Deepak Ailani[3], Owen Randlett[1], Ji-Yoon Kim[1], Maxim Nikitchenko[1], Armin Bahl[1], Chao-Tsung Yang[4], Misha B Ahrens[4], Koichi Kawakami[3], Florian Engert[1], Sam Kunes[1]*

[1]Department of Molecular and Cell Biology, Harvard University, Cambridge, United States; [2]Program in Neuroscience, Harvard University, Boston, United States; [3]Laboratory of Molecular and Developmental Biology, National Institute of Genetics, Department of Genetics, SOKENDAI (The Graduate University for Advanced Studies), Mishima, Japan; [4]Howard Hughes Medical Institute, Janelia Research Campus, Ashburn, United States

**Abstract** Medial and lateral hypothalamic loci are known to suppress and enhance appetite, respectively, but the dynamics and functional significance of their interaction have yet to be explored. Here we report that, in larval zebrafish, primarily serotonergic neurons of the ventromedial caudal hypothalamus (cH) become increasingly active during food deprivation, whereas activity in the lateral hypothalamus (LH) is reduced. Exposure to food sensory and consummatory cues reverses the activity patterns of these two nuclei, consistent with their representation of opposing internal hunger states. Baseline activity is restored as food-deprived animals return to satiety via voracious feeding. The antagonistic relationship and functional importance of cH and LH activity patterns were confirmed by targeted stimulation and ablation of cH neurons. Collectively, the data allow us to propose a model in which these hypothalamic nuclei regulate different phases of hunger and satiety and coordinate energy balance via antagonistic control of distinct behavioral outputs.

**\*For correspondence:**
carolinewee@gmail.com (CLW);
kunes@fas.harvard.edu (SK)

†These authors contributed equally to this work

**Present address:** ‡Institute of Molecular and Cell Biology, A*STAR, Singapore, Singapore

**Competing interests:** The authors declare that no competing interests exist.

## Introduction

The regulated intake of food based on caloric needs is a fundamental homeostatically controlled process that is essential for health and survival. The hypothalamus is a highly conserved central convergence point for the neural and biochemical pathways that underlie this regulatory mechanism. Early studies demonstrated by way of electrical stimulation or lesions that specific hypothalamic regions play important roles in the regulation of appetite. For example, while stimulation of ventromedial hypothalamic loci in rodents and cats reduced feeding, activation of more lateral hypothalamic loci increased both hunting behavior and food intake (*Anand and Brobeck, 1951*; *Brobeck et al., 1956*; *Delgado and Anand, 1952*; *Krasne, 1962*). Conversely, lateral hypothalamic lesions were found to reduce feeding to the point of starvation, whereas medial hypothalamic lesions resulted in overeating (*Anand and Brobeck, 1951*; *Hoebel, 1965*; *Teitelbaum and Epstein, 1962*). Thus, the lateral and medial hypothalamic regions came to be regarded as 'hunger' and 'satiety' centers, respectively.

Recent experiments employing optical and electrophysiological methods have lent support to these early studies. For example, GABAergic neurons in the lateral hypothalamus were observed to be activated during feeding and essential for enhanced food intake during hunger (*Jennings et al., 2015*; *Stuber and Wise, 2016*). However, these experiments have examined only subsets of hypothalamic neurons; their activity patterns and function within the context of the entire network remain

**eLife digest** How soon after a meal do you start feeling hungry again? The answer depends on a complex set of processes within the brain that regulate appetite. A key player in these processes is the hypothalamus, a small structure at the base of the brain. The hypothalamus consists of many different subregions, some of which are responsible for increasing or decreasing hunger.

Wee, Song et al. now show how two of these subregions interact to regulate appetite and feeding, by studying them in hungry zebrafish larvae. The brains of zebrafish have many features in common with the brains of mammals, but they are smaller and transparent, which makes them easier to study. Wee, Song et al. show that as larvae become hungry, an area called the caudal hypothalamus increases its activity. But when the larvae find food and start feeding, activity in this area falls sharply. It then remains low while the hungry larvae eat as much as possible. Eventually the larvae become full and start eating more slowly. As they do so, the activity of the caudal hypothalamus goes back to normal levels.

While this is happening, activity in a different area called the lateral hypothalamus shows the opposite pattern. It has low activity in hungry larvae, which increases when food becomes available and feeding begins. When the larvae finally reduce their rate of feeding, the activity in the lateral hypothalamus drops back down. The authors posit that by inhibiting each other's activity, the caudal and lateral hypothalamus work together to ensure that animals search for food when necessary, but switch to feeding behavior when food becomes available.

Serotonin – which is produced by the caudal hypothalamus – and drugs that act like it have been proposed to suppress appetite, but they have varied and complex effects on food intake and weight gain. By showing that activity in the caudal hypothalamus changes depending on whether food is present, the current findings may provide insights into this complexity. More generally, they show that mapping the circuits that regulate appetite and feeding in simple organisms could help us understand the same processes in humans.

unknown. This limited view hampers our understanding of the dynamical interactions between the ensemble of brain circuits thought to be important for the initiation, maintenance and termination of food consumption (*Sternson and Eiselt, 2017*).

Here, we leverage the small and optically accessible larval zebrafish to identify modulatory regions central to the control of appetite and to shed light on their specific roles and dynamical activity patterns in relation to behavior. Using pERK-based brain-wide activity mapping (*Randlett et al., 2015*), we first identified neuronal populations that display differential neural activity under conditions that would yield hunger and satiety. We show that lateral and medial hypothalamic regions have anti-correlated activity patterns during food deprivation, and voracious or steady state feeding. Next, through a combination of calcium imaging, optogenetics and ablation analysis, we show that serotonergic neurons in the caudal periventricular zone of the medial hypothalamus (cH) are state-dependent regulators of feeding behavior, most likely via their modulation of lateral hypothalamic activity. These results allow us to propose a model where mutually antagonistic brain states regulate energy balance by encoding distinct signals for different facets of appetite control.

## Results

### Whole brain activity mapping of appetite-regulating regions

Larval zebrafish hunt prey such as paramecia through a sequence of motor actions that has been considered a hardwired reflex response to external prey stimuli (*Bianco et al., 2011*; *Semmelhack et al., 2015*; *Trivedi and Bollmann, 2013*). Only recently has evidence emerged that this behavior is flexibly modulated by satiation state (*Filosa et al., 2016*; *Jordi et al., 2015*; *Jordi et al., 2018*) and that larvae at 7 **d**ays **p**ost-**f**ertilization (dpf) display enhanced hunting and enhanced food intake after a period of food deprivation. A robust readout of food intake in larval zebrafish was obtained both by the ingestion of fluorescently-labeled paramecia and by behavioral analysis, using protocols adapted for this study (*Johnson et al., 2019*; *Jordi et al., 2015*; *Jordi et al., 2018*; *Shimada et al., 2012*). A 2 hr period of food deprivation robustly enhances

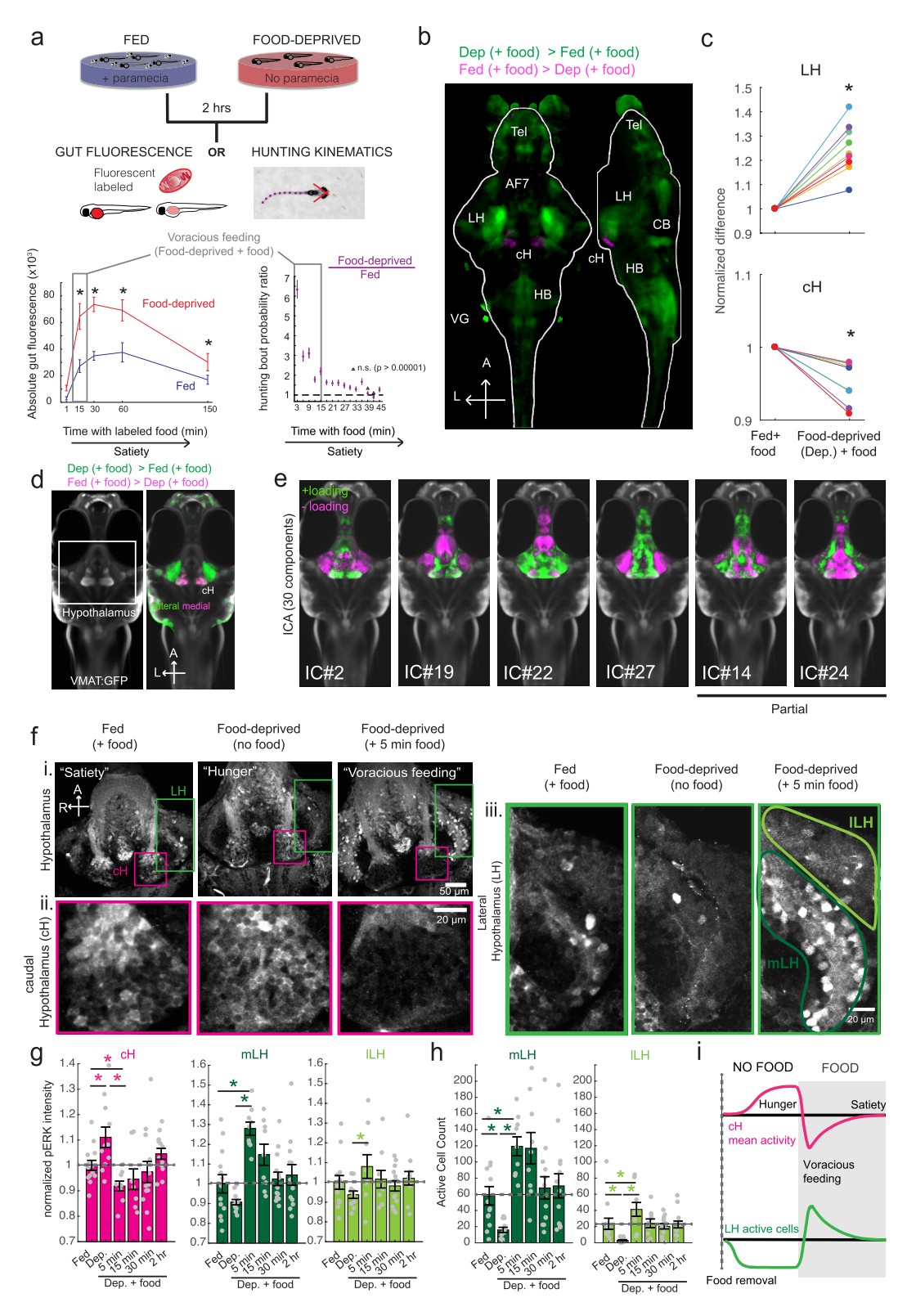

**Figure 1.** Whole brain activity mapping reveals anti-correlated hypothalamic regions. (a) Top: The protocols used to quantify feeding behavior in larval zebrafish. At 7 or 8 dpf, larvae were either food-deprived for 2 hr, or fed with excess paramecia for this duration. After 2 hr (2–4 hr in the case of behavioral imaging), they were subject to a quick wash, followed either by: 1) addition of excess fluorescently-labeled paramecia (left), 2) high-resolution behavioral imaging (right; see *Johnson et al., 2019*, and Materials and methods). Gut fluorescence is both cumulative and diminished by

*Figure 1 continued on next page*

*Figure 1 continued*

digestion (*Jordi et al., 2015*) and so lags the dynamics of hunting behavior. Bottom left: Gut fluorescence measurements of food-deprived (red) or fed (blue) fish as a function of duration of feeding labeled paramecia. Groups of fed or food-deprived larvae were fixed at the indicated time points (fed: n = 7/18/19/17/17 fish, food-deprived: n = 8/23/20/14/15 fish). Food-deprived fish had significantly higher gut fluorescence than fed fish overall (p = $7.5859 \times 10^{-10}$, Two-way ANOVA, asterisk indicates corrected p-values<0.05. Bottom right: The probability of performing a hunting-related swim bout across fed and food-deprived fish groups in 3 min time bins over 45 min. Error bars represent 90% confidence intervals. For all bins except those indicated with triangles, the null hypothesis that initial feeding condition has no effect on hunting-bout probability is rejected (p<0.00001, Fisher's Exact Test comparing binomial probability distributions per bin). Fed: n = 85655 bouts from 73 fish; Food-deprived: n = 75357 bouts from 57 fish. Since the rate of food intake and hunting behavior was highest in the first 15 min (voracious feeding phase, gray boxes), we chose this time point for subsequent MAP-mapping experiments. (b) Brain-wide activity mapping of food-deprived (Dep.) fish exposed to food for 15 min, with subtraction of activity in continuously fed (Fed) fish. Data from nine experiments were combined to generate this difference map based on anti-pERK staining fluorescence. Relative activation from feeding after food deprivation yields activated regions including the telencephalon (Tel), Arborization field 7 (AF7), cerebellum (CB), hindbrain (HB), Vagal ganglion (VG) and lateral lobe of the intermediate hypothalamus (LH). Reduced activity was observed in the caudal hypothalamus (cH) and some areas of the telencephalon. Scale bar = 100 µm. Also see *Video 1*. (c) ROI-specific pixel intensity analysis of LH and cH regions in nine independent MAP-mapping experiments (20–30 fish per treatment per experiment). The cH or LH ROI intensities of each individual fish was normalized to the mean cH or LH ROI intensity of all fed fish. Food-deprived fish consistently displayed higher LH and lower cH pERK fluorescence after the onset of feeding (p = 0.0019 for both cH and LH, one-tailed Wilcoxon signed-rank test). (d) Z-projection of same MAP-map as described in (b) in planes revealing the hypothalamus (right panel), where lateral regions (e.g. lateral hypothalamus, LH) display strong relative activation and medial regions (e.g. caudal hypothalamus, cH) display reduced activity in when food-deprived animals were fed for 15 min. The map is overlaid onto a stack for the transgenic line *Tg(VMAT:GFP)* (left panel) to localize the cH region. (e) Six examples of independent component analysis (ICA) maps. Voxels for each recovered independent component (IC) are shown as maximum projections, with intensity proportional to the z-score of the loadings of the ICA signal. These ICs, along with others (22/30) highlight LH and cH regions of opposite loadings, suggesting they may be included in a network that displays anti-correlated activity patterns between the cH and LH. A subset of these ICs (e.g. #14 and #24) only showed partial anti-correlation between the cH and the LH. All ICs are shown in *Figure 1—figure supplement 3*. Positive (+) loading and Negative (-) loadings (z-score values of IC signals) are reflected in green and magenta, respectively. (f) Confocal micrographs of anti-pERK antibody stained brains from animals that were continuously fed (panel (i), left), food-deprived for 2 hr (panel (i), center) and fed for 5 min after food deprivation (panel (i), right). cH (ii) and LH (iii) insets are shown at higher magnification on the bottom and right side respectively. The lateral hypothalamus is shown with subdivisions *lateral lateral hypothalamus* (lLH) and *medial lateral hypothalamus* (mLH). (i) scale bar: 50 µm; (ii) and (iii) scale bar: 20 µm. Fish are mounted ventral side up. (g) Quantification of cH and LH activities by normalized anti-pERK fluorescence intensity averaging. The normalized anti-pERK staining intensity for each region (ROI) was obtained by dividing the anti-pERK fluorescence from each fish (in all experimental groups) by the average anti-pERK fluorescence for the same ROI of continuously fed fish. Quantitative analysis performed on fish in six independent conditions (n = 13/11/9/9/13/12). Normalized anti-pERK fluorescence intensity (cH/mLH/lLH): Fed vs Dep. (p = 0.016/0.12/0.11), Dep. vs Dep. + 5 min food (p = $3.1 \times 10^{-4}$/$9.9 \times 10^{-5}$/0.020), Fed vs Dep. + 5 min food (p = 0.0097/$8.5 \times 10^{-4}$/0.11). Asterisks denote p<0.05, one-tailed Wilcoxon rank-sum test. (h) The active cell count metric (bottom panels) was determined as described in *Figure 1—figure supplement 4* by a thresholding protocol to isolate and count individual pERK-positive cells within a z-stack. This approach could be reliably performed for areas of sparse active cells (e.g. mLH and lLH) but not where individually labeled pERK-positive neurons are not well separated (such as the cH). Active cell count (mLH/lLH): Fed vs Dep. (p = 0.001/0.0038), Dep. vs Dep. + 5 min food (p = $9.7 \times 10^{-5}$/ $1.3 \times 10^{-5}$), Fed vs Dep. + 5 min food (p = 0.0038/0.048). Asterisks denote p<0.05, one-tailed Wilcoxon rank-sum test. (i) Schematic of inferred cH and LH activity in relation to feeding behavior. Note that, based on data in *Figure 2*, the LH active cell count appears to decline more rapidly than the rise in cH activity (based on cH average fluorescence intensity). Data plotted in *Figure 1* are provided in *Figure 1—source data 1*.

The online version of this article includes the following source data and figure supplement(s) for figure 1:

**Source data 1.** Source data for plots displayed in *Figure 1a, c, g and h*.
**Figure supplement 1.** Anatomical characterization of intermediate hypothalamus expression of appetite related peptides.
**Figure supplement 2.** Characterization of neuronal transmitter types in the zebrafish lateral hypothalamus.
**Figure supplement 3.** All 30 independent components extracted from ICA analysis.
**Figure supplement 4.** Automated quantification of pERK-positive (active) cells.
**Figure supplement 5.** Food deprivation-induced activity in caudal hypothalamus monoaminergic neurons.

subsequent food intake (*Figure 1a*). Up to 15 min after the presentation of prey, food-deprived animals display a strong upregulation of hunting and prey intake relative to fish that have continuous access to food (referred to as *fed fish*; *Figure 1a*), on the basis of fluorescent food ingestion (left panel, *Figure 1a*) and hunting bouts (right panel, *Figure 1a*). We refer to this behavior as 'voracious feeding'. Finally, as the fish consume food, their rate of food intake declines to that of continuously fed fish (*Figure 1a*). These behaviors likely represent internal states that are commonly referred to as hunger and satiety, and reflect the animal's underlying caloric or metabolic needs.

As a first step toward understanding the homeostatic control of feeding in this simple vertebrate system, we employed whole-brain neuronal activity mapping via phosphorylated ERK visualization in post-fixed animals (MAP-mapping; *Randlett et al., 2015*). Whole brain confocal image datasets of phospho-ERK expression were gathered from animals sacrificed after 15 min of voracious feeding

that followed a 2 hr period of food deprivation. For comparison, image sets were also gathered from animals that had been fed continuously (*fed fish*). The image volumes were registered to a standardized brain atlas. A difference map (*Figure 1b*) reveals significant specific differences in neural activity when comparing voracious feeding with continuous feeding (*Figure 1b–d*, *Video 1*, *Supplementary files 1–2*). Since both experimental groups experienced the same sensory stimuli (i.e. exposure to paramecia) prior to sacrifice, differences in brain activity should primarily reflect the animals' internal states, which could include manifestations of an altered sensitivity to food cues, activity related to hunting and prey capture, or the motivational history resulting from food deprivation. Indeed, multiple sensorimotor loci related to hunting showed enhanced activity during feeding that followed the food-deprived condition, consistent with the increased feeding behavior observed in food-deprived animals. These loci included the retinal Arborization Fields (AFs; optic tectum and AF7), pretectum, as well as downstream hindbrain loci, such as reticulospinal and oculomotor neurons, all of which are known to be engaged during prey capture behavior (*Bianco and Engert, 2015*; *Muto et al., 2017*; *Semmelhack et al., 2015*). In addition, enhanced activity was observed in the cerebellum, inferior olive, vagal sensory and motor neurons, area postrema and locus coeruleus,

areas that have been implicated in feeding regulation and behavior (*Ahima and Antwi, 2008*; *Ammar et al., 2001*; *Dockray, 2009*; *Zhu and Wang, 2008*).

We focused our attention on brain areas likely to be involved in motivational states related to feeding. These included an area of particularly strong differential activity in the lateral region of the intermediate hypothalamus (Lateral Hypothalamus, LH; *Figure 1b–d*), which has recently been identified as part of the feeding pathway in larval zebrafish (*Muto et al., 2017*) and whose mammalian analog has been strongly implicated in appetite control (*Sternson and Eiselt, 2017*). However, the zebrafish LH, unlike its mammalian counterpart, does not harbor melanin-concentrating hormone (MCH)-positive, orexin (hypocretin)-positive neurons, or other major feeding-related peptides (*Figure 1—figure supplements 1* and *2*). We therefore characterized the expression of multiple appetite-related neuromodulators (AgRP, MSH, CART, NPY, MCH, Orexin) and found that they are instead expressed in nearby areas of the hypothalamus (*Figure 1—figure supplement 1*). The zebrafish LH region does however contain glutamatergic and GABAergic cell types (*Figure 1—figure supplement 2*); these non-peptidergic LH cell types have been shown in rodents to be important for the regulation of feeding (*Jennings et al., 2015*; *Stuber and Wise, 2016*).

Among areas that showed relatively decreased neural activity upon feeding food-deprived animals, the most significant was the adjacent caudal hypothalamus (cH), which contains monoaminergic neurons – mainly serotonergic and dopaminergic cells, with a small fraction of histaminergic cells (*Chen et al., 2016*; *Kaslin and Panula, 2001*; *Lillesaar, 2011*). Indeed, in all of nine independent MAP-mapping experiments, activity was reduced in the cH and

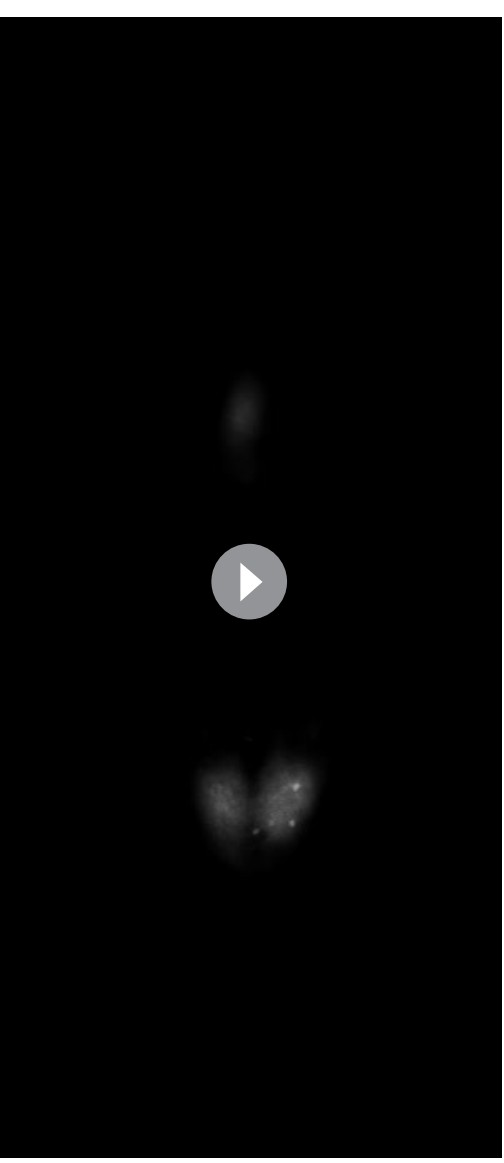

**Video 1.** Z-stack (dorsal to ventral) of brain activity map shown in *Figure 1b*.
https://elifesciences.org/articles/43775#video1

increased in the LH within 15 min of food presentation (*Figure 1c*). The evident inverse relationship between LH and cH neural activity is supported by independent component analysis (*Randlett et al., 2015*), which was applied to feeding-related MAP-mapping data (*Figure 1e*, *Figure 1—figure supplement 3*). Multiple components were uncovered in which cH and LH activities were strongly anti-correlated. These results led us to hypothesize that the lateral and caudal hypothalamic regions form a functionally interconnected network with opposing activity patterns.

## Cellular dissection of hypothalamus neural activity reveals modulation by satiation state

To probe these neural activity changes at higher resolution, we performed anti-pERK antibody staining on isolated brains and examined the hypothalamus in time course experiments spanning a period of food deprivation and subsequent feeding (*Figure 1f–h*, *Figure 2*). We quantified the mean anti-pERK fluorescence within a region-of-interest (ROI; *Figure 1g*) as well as the number of active cells or cell clusters (*Figure 1h*; *Figure 1—figure supplement 4*). These two metrics were employed because the high density of pERK-positive cells in the cH of food-deprived animals made high-throughput quantitation of active cells unreliable, whereas use of this metric in areas of sparse activity (e.g. mLH and lLH) yielded better differential sensitivity than ROI averaging. Using these respective metrics, we observed that mean fluorescence in the cH was significantly increased in food-deprived fish, while the number of active neurons in the medial and lateral lobes of the LH (mLH and lLH, respectively) was relatively low (*Figure 1f–h*). Within the cH, enhanced pERK activity during food deprivation was most prevalent in serotonergic neurons, but also present in a smaller proportion of dopaminergic neurons (*Figure 1—figure supplement 5*, *Videos 2* and *3*).

During the period of voracious feeding that followed food deprivation, the pERK-reported activity of cH neurons fell dramatically to a level significantly below that observed in continuously fed fish (*Figure 1f–h*). This characteristically low cH activity level coincided with a large increase in LH activity, measured by either mean anti-pERK fluorescence or by measurement of the number of individually active neurons, that lasted throughout the period of voracious feeding. Thereafter, as feeding continued at a more moderate pace, and the rate of food ingestion declined, LH neuronal activity likewise declined (especially for lLH neurons; *Figure 1h*). Reciprocally, cH activity slowly increased back towards baseline levels. After 30 min of feeding, neural activity in both the cH and LH had mostly converged to the baseline level observed for continuously fed fish, consistent with the time course of hunting behavior reduction (*Figure 1a*, right panel). Thus these cH and LH populations displayed anti-correlated activity over time frames that spanned a progression of distinct behaviors associated with food deprivation, voracious feeding and a gradual return to apparent satiety (*Figure 1i*).

## Satiation state influences the responses of cH and LH populations to food

To more closely align the activity patterns of cH and LH neuronal populations with feeding behavior, we examined these areas after a 30 min (i.e. short) or 2–4 hr (i.e. long) period of food deprivation, with or without a subsequent period of feeding (*Figure 2*, *Figure 2—figure supplement 1*). Following food removal, cH activity increased, with an especially large anti-pERK average fluorescence intensity increase after 2 hrs of food deprivation (*Figure 2a–b*). In contrast to the cH, food removal quickly reduced the frequency of active mLH and lLH neurons (*Figure 2a,c*). Despite the reduction in LH active cell count over food deprivation, there were no obvious changes in mean LH anti-pERK fluorescence over the course of food deprivation (*Figure 2b*). This is because there are few active LH cells in continuously fed and food-deprived fish, thus their overall contribution to the fluorescence average of the mLH and lLH regions of interest is small.

Notably, the addition of prey (paramecia) rapidly reversed the food deprivation- induced patterns of cH and LH neural activity, with an amplitude of change that was correlated with the length of food deprivation (*Figure 2a–c*, *Figure 2—figure supplement 1d–e*). Fish that had been food-deprived for longer periods (2 hr or 4 hr) displayed a greater increase in the number of active LH neurons compared to feeding animals that had been food-deprived for only 30 min (*Figure 2a–c*; *Figure 2—figure supplement 1d–e*). Likewise, the reduction in cH activity after food presentation was greater when it followed a longer period of prior deprivation (*Figure 2a–b*; *Figure 2—figure*

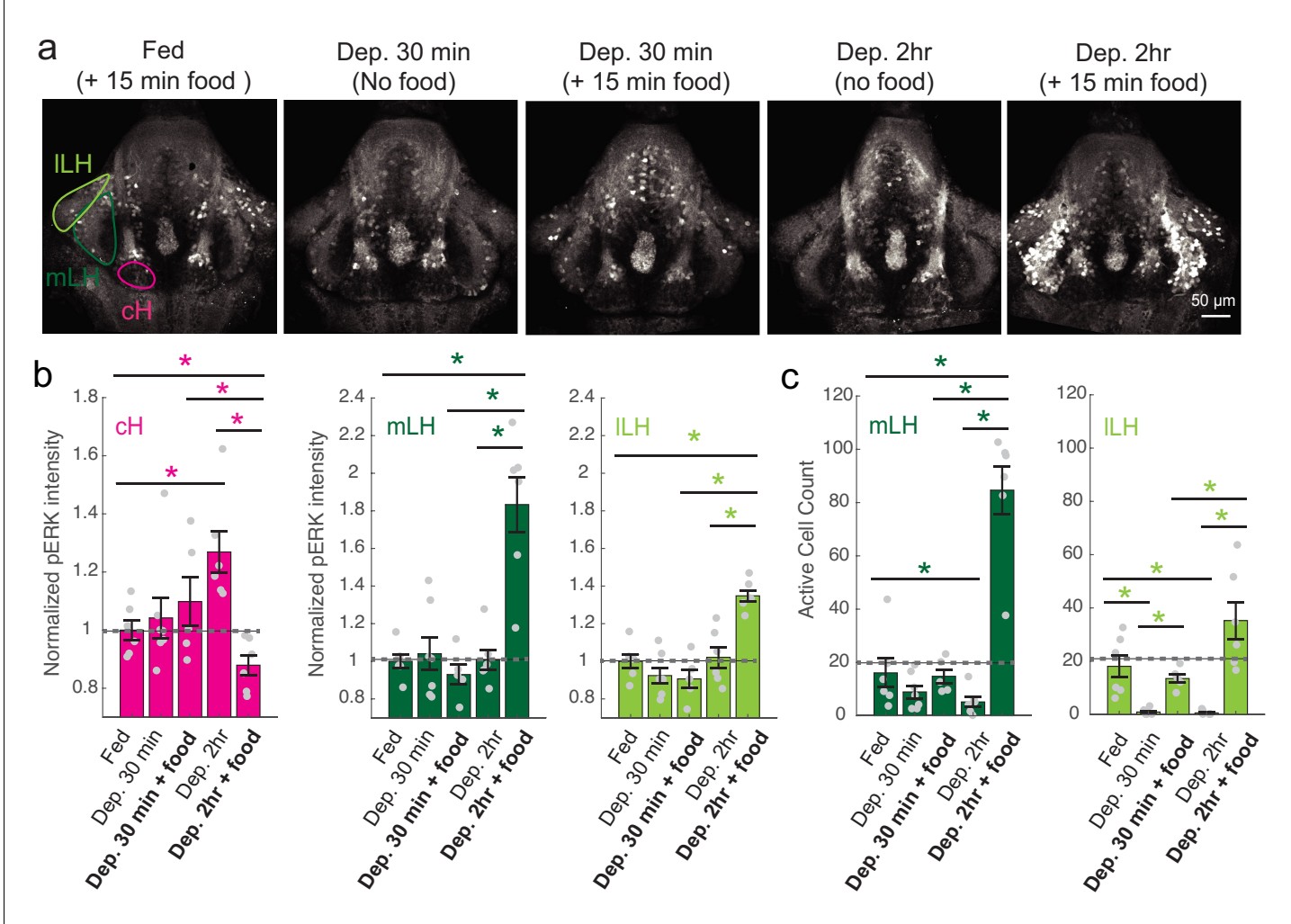

**Figure 2.** cH and LH activities are modulated by food and satiation state. (a) Representative images showing that cH, mLH and lLH activities in the presence and absence of food vary with the extent of food deprivation (dataset quantified in b and c). (b) Normalized pERK average fluorescence intensity in cH significantly increases with food deprivation, and is significantly reduced when food is presented to food-deprived fish. Normalized mLH and lLH pERK average fluorescence intensity does not change significantly during food deprivation and strongly increases during voracious feeding (Dep. 2 hr + 15 min food). Asterisks denote p<0.05. Normalized pERK intensity (cH/mLH/lLH): Fed vs Dep. 30 min (p = 0.53/0.47/0.15), Fed vs Dep. 2 hr (p = 0.0022/0.41/0.59), Dep. 30 min + food vs Dep. 2 hr + food (p = 0.041/0.0022/0.0022), Dep. 30 min vs Dep. 30 min + food (p = 0.62/0.73/0.62), Dep. 2 hr vs Dep. 2 hr + food (p = 0.0022/0.0011/0.0022), Fed vs Dep. 2 hr + food (0.047/0.0011/0.0011). Anti-pERK staining fluorescence was averaged over each entire region of interest (i.e. cH, mLH and lLH; see Materials and methods for details). The normalized anti-pERK staining intensity for each region (ROI) was obtained by dividing the anti-pERK fluorescence from each fish (in all experimental groups) by the average anti-pERK fluorescence for the same ROI of continuously fed fish. (c) The number of active mLH and lLH cells declines within 30 min of food deprivation, and is significantly enhanced during feeding, particularly after a longer period of food deprivation. Active cell count (mLH/lLH): Fed vs Dep. 30 min (p = 0.155/5.8 × 10$^{-4}$), Fed vs Dep. 2 hr (p = 0.047/0.011), Dep. 30 min + food vs Dep. 2 hr + food (p = 0.0022/0.0043), Dep. 30 min vs Dep. 30 min + food (p = 0.07/0.013), Dep. 2 hr vs Dep. 2 hr + food (p = 0.0011/0.0011), Fed vs Dep. 2 hr + food (p = 0.0022/0.07), n = 6/7/5/6/6 fish, one-tailed Wilcoxon rank-sum test.

Data plotted in *Figure 2* are provided in *Figure 2—source data 1*.

The online version of this article includes the following source data and figure supplement(s) for figure 2:

**Source data 1.** Source data for plots displayed in *Figure 2b-c*.

**Figure supplement 1.** Modulation of cH, mLH and lLH activity in relation to feeding.

*supplement 1d*). In general, the presence of highly active neurons in the LH was correlated with higher food consumption (as measured by gut fluorescence, *Figure 2—figure supplement 1a–e*).

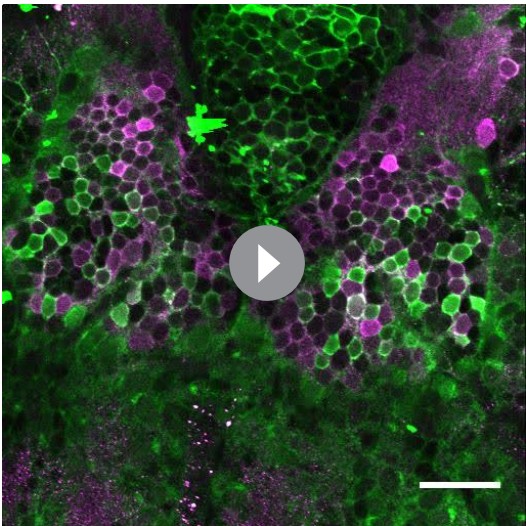

**Video 2.** Z-stack (ventral to dorsal) of anti-5-HT (green) and anti-pERK (magenta) staining in food-deprived fish. Scale bar = 20 µm.

https://elifesciences.org/articles/43775#video2

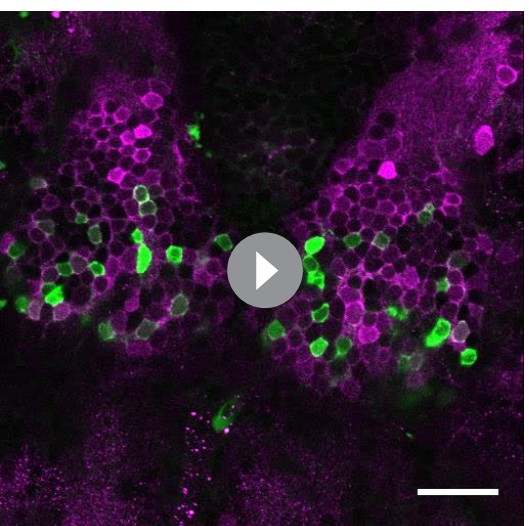

**Video 3.** Z-stack (ventral to dorsal) of TH2:GCaMP5 transgene expression (green) and anti-pERK (magenta) staining in the same food-deprived fish as in **Video 2**. Scale bar = 20 µm.

https://elifesciences.org/articles/43775#video3

## Caudal and lateral hypothalamic responses to food sensory cues are anti-correlated over short timescales

We next set out to characterize acute effects of food sensory cues on both the cH and LH, and also to analyze in more detail the apparent negative activity relationship between these two nuclei. Such analyses require higher temporal resolution than afforded by anti-pERK staining analysis, thus we switched to in vivo calcium imaging of the cH and LH in live animals (*Figure 3*). To that end, two transgenic Gal4 drivers, *Tg(116A:Gal4)* and *Tg(76A:Gal4)*, were combined to express GCaMP6s (*Tg(UAS:GCaMP6s)*) in neuronal subsets of both the cH and LH (*Figure 3—figure supplements 1–2*). The 116A:Gal4 transgene drives expression mainly in serotonergic neurons of the cH (88.9 ± 0.8% 5-HT positive) and paraventricular organ (PVO; *Figure 3—figure supplement 1*), whereas 76A:Gal4 drives expression in a large proportion of LH cells (*Figure 3—figure supplement 2*; *Muto et al., 2017*).

Using these transgenic animals, we examined calcium dynamics in the cH and LH regions in tethered animals during the controlled presentation of prey stimuli (*Figure 3a*). In these experiments, live paramecia were released in a puff of water in the vicinity of the immobilized fish, which can neither hunt nor ingest prey. Consistent with the results of anti-pERK analysis of post-fixed brains (*Figures 1* and *2*), activity in the mLH and lLH regions was increased and cH activity quickly reduced, in fact within seconds of paramecia release (*Figure 3b,d*). Neurons in all three hypothalamic loci also responded to water flow alone, but these responses were significantly less than those elicited by paramecia (*Figure 3b,d,e*). These prey-induced changes in activity were particularly striking for the mLH region, which displayed both a strongly enhanced calcium spike frequency and spike amplitude upon the introduction of prey. Thus, prey sensory cues, even in the absence of hunting or prey ingestion, strongly and differentially regulate neuronal activity in the caudal and lateral hypothalamus.

The activities of cH and LH neurons also appeared remarkably anti-correlated; both spontaneous and prey-induced fluctuations in one population were accompanied by corresponding opposing activity changes in the other (*Figure 3b–c*). This observation was supported by cross-correlation analysis between cH, mLH and lLH voxels (*Figure 3f*), which revealed high correlation within the same hypothalamic region (red color), and anti-correlation between cH and LH regions (blue color). Further, lLH voxels showed more spatial heterogeneity than mLH voxels (*Figure 3f*), though a small cluster of cells at the most-anterior part of the lLH was most consistently anti-correlated with cH activity (Fish C and D, black arrowheads). When ranked

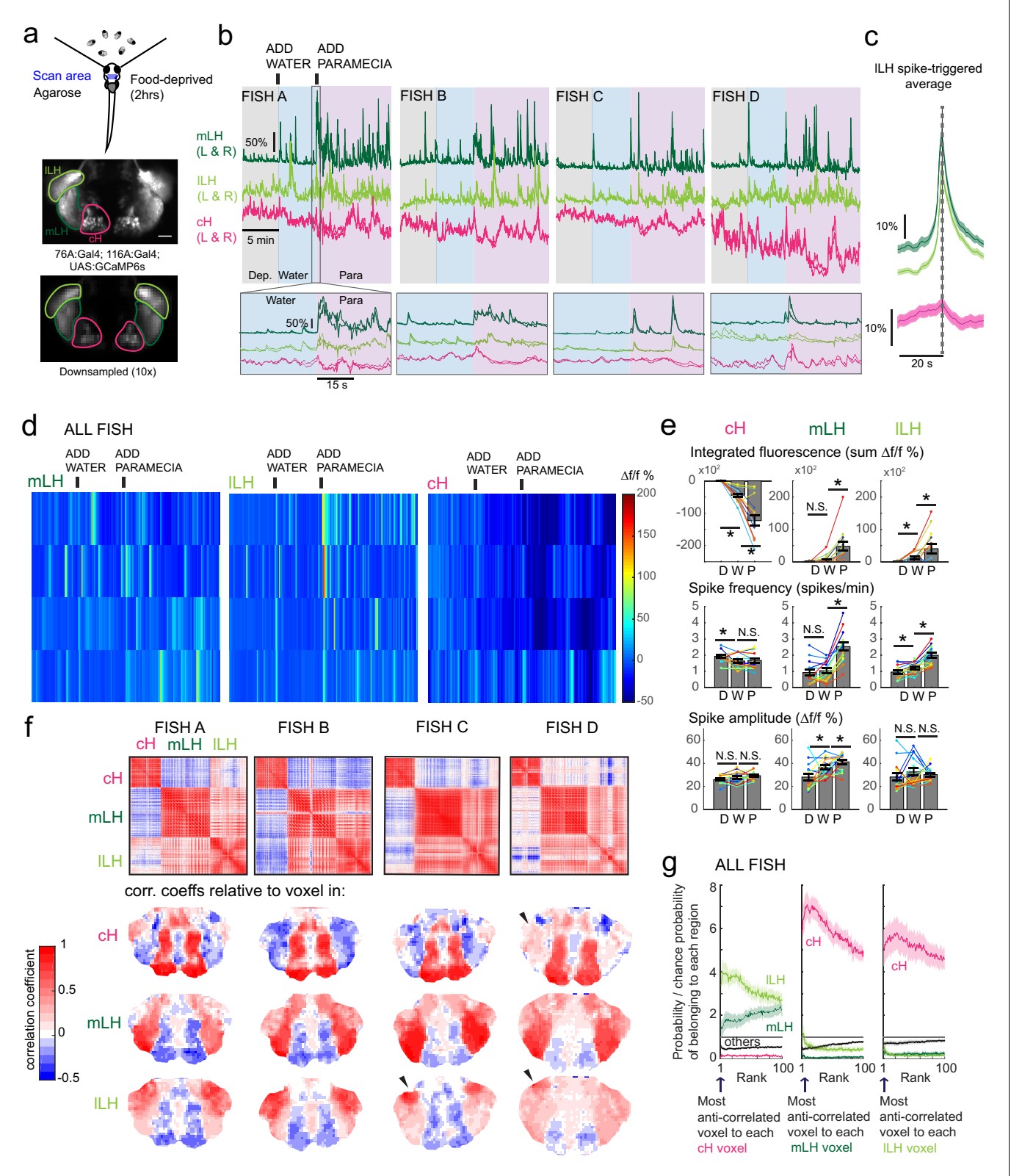

**Figure 3.** Caudal and lateral hypothalamic responses to prey sensory cues are anti-correlated over short timescales. (**a**) Top: Transgenic fish (2 hr food-deprived) with GCaMP6s expressed in cH and LH neurons were paralyzed, tethered in agarose with their eyes and nostrils freed and exposed to live paramecia (prey), as described in Materials and methods. Top image: GCaMP expression in the cH and LH driven by two transgenic lines, *Tg(116A: Gal4) and Tg(76A:Gal4)* respectively. Bottom image: Downsampled image stack used for analysis in (**f**). (**b**) Top: Mean calcium activity (Δf/f) from

*Figure 3 continued on next page*

Figure 3 continued

respective hypothalamic ROIs (shown in (a)) from four individual fish during a baseline food-deprived period (Dep.), exposure to water alone (Water), and a dense water drop of paramecia (Para). Traces from left and right hypothalamic lobes of the same animal are overlain, revealing a high degree of correlated activity on opposite sides of the midline. Paramecia presentation increases activity in the LH and reduces activity in the cH, revealing opposing activity on short timescales. Bottom: $\Delta f/f$ traces within area marked by gray box (top), displayed at higher magnification. An increase in LH activity and corresponding reduction in cH activity is observable within seconds of paramecia presentation, except for fish D in which maximal responses only occur after a few minutes (beyond the displayed time window). (c) Average $\Delta f/f$ triggered on lLH calcium spikes (left and right lobes averaged) shows a mean corresponding reduction in cH activity (n = 159 lLH spikes extracted from mean $\Delta f/f$ traces from 14 fish across the entire duration of the experiment). (d) Raster plots showing mean calcium activity from the hypothalamic lobes (left and right lobes averaged) of 14 fish before and after presentation of water alone and water with paramecia. (e) Quantification of integrated fluorescence (sum $\Delta f/f$ %), calcium spike frequency (spikes/min) and calcium spike amplitude ($\Delta f/f$ %) per fish across experimental epochs (300 s food-deprived baseline (D), 300 s after water (W) delivery or 600 s after paramecia delivery (P)). Each colored line represents data from an individual fish (left and right lobes averaged). Water alone was sufficient to significantly reduce cH integrated fluorescence (p = $6.1\times10^{-5}$) and spike frequency (p = 0.0127) but not spike amplitude (p = 0.9324). Water alone was similarly sufficient to increase lLH integrated fluorescence (p = 0.029) and spike frequency (p = 0.0098) but not spike amplitude (p = 0.13). Conversely, water alone was not sufficient to significantly modulate mLH integrated fluorescence (p = 0.48) or spike frequency (p = 0.20), but was sufficient to increase spike amplitude (p = 0.039). Paramecia delivery significantly increased mLH and lLH integrated fluorescence (mLH, p = $1.2\times10^{-4}$; lLH, p = 0.045) and spike frequency (mLH, p = $6.1\times10^{-5}$; lLH, $6.1 \times 10^{-4}$), while only significantly increasing mLH spike amplitude (mLH, p = 0.045; lLH, p = 0.43), relative to water delivery. In contrast, paramecia delivery significantly reduced cH integrated fluorescence relative to water delivery alone (p = $3.1\times10^{-4}$), but not spike frequency (p = 0.52) nor spike amplitude (p = 0.85). Asterisks denote p<0.05, one-tailed Wilcoxon signed-rank test. (f) Top: Cross-correlogram of hypothalamic cell-sized voxels (cells and/or neuropil from downsampled image stacks, see (a)) from four fish. The cH and LH voxels were mostly anti-correlated, whereas voxels within each cluster displayed correlated activity. Black arrowheads indicate region of lLH that appears to be most anti-correlated with the cH. Bottom: Correlation coefficients of other hypothalamic voxels relative to a selected voxel with the cH, mLH or lLH. See color key for numerical translation of color maps. (g) Summary of data from 14 fish, showing the probability of the $n^{th}$ most anti-correlated voxel belonging to each of the other regions (cH, mLH or lLH), normalized to chance probability (gray line) of belonging to each region (i.e. the fraction of all voxels occupied by each region). For example, if we consider all the voxels within the cH, there is a four-fold probability relative to chance of their most anti-correlated voxels (Rank = 1) being part of the lLH.

The online version of this article includes the following figure supplement(s) for figure 3:

**Figure supplement 1.** Characterization of the 116A:Gal4 line.

**Figure supplement 2.** Overlap of 116A:Gal4 and 76A:Gal4 driven reporter expression with hypothalamic activity under conditions of food deprivation and feeding.

**Figure supplement 3.** Calcium imaging of the cH and LH over food deprivation.

according to their degrees of anti-correlation with voxels from other lobes, the cH and lLH displayed the greatest anti-correlation (*Figure 3g*). Overall, these results indicate that cH and LH neurons display generally anti-correlated activities over short timescales, in addition to the anti-correlation observed over longer epochs reflecting motivational states imposed by food deprivation and feeding.

In addition to these studies over short timescales, we also analyzed live imaging traces that spanned extended time periods (up to 2 hr) of food deprivation (*Figure 3—figure supplement 3a*). This long-term imaging resulted in some confounding modulation of baseline fluorescence over these timescales (*Figure 3—figure supplement 3a*, particularly lLH trace), that do not necessarily reflect changes in neural firing (*Berridge, 1998*; *Verkhratsky, 2005*) and may well be related to modified internal states caused by tethering and immobilization. Nonetheless, we observed significantly higher calcium spike frequencies and amplitudes in the cH as compared to LH regions over the course of food deprivation (*Figure 3—figure supplement 3a,c–d*), activity patterns that were the opposite of those observed for these regions when prey was presented (*Figure 3b,e*). For example, the calcium spike amplitude and frequency of the cH region were many-fold greater than those observed in the mLH region during food deprivation (*Figure 3—figure supplement 3d*), whereas after prey presentation, these relative activities were reversed, with the mLH displaying significantly greater spike amplitude and frequency than the cH (*Figure 3b,e*). Likewise, lLH calcium spike frequency is significantly lower than the cH during food deprivation, but increases significantly after prey presentation (*Figure 3—figure supplement 3d*, *Figure 3e*). Thus, the cH is more active over food deprivation, and the LH under conditions where food is present.

## Separation of cH and LH neuronal activities associated with prey detection and ingestion

We next sought to characterize the responses of hypothalamic regions to prey ingestion, as opposed to the mere detection of prey. To distinguish between the consequences of sensory and consummatory inputs, we compared neural activities in food-deprived fish exposed to paramecia or artemia. Artemia are live prey commonly fed to adult zebrafish and are actively hunted by fish at all stages, including larvae (*Figure 4a*, *Video 4*). Thus, artemia provide sensory inputs that elicit hunting behavior in larval animals. They are however too large to be swallowed and consumed by larvae. Hence, the comparison between these two types of prey dissociates neural activity triggered by prey detection and hunting from that of food ingestion.

Prey ingestion can only occur in freely behaving animals and thus we needed to return to pERK-based activity mapping in post-fixed animals for our analysis. We found that artemia exposure caused significant increases in both mLH and lLH activity, whereas little change was detected in cH neurons (*Figure 4a–c*). Exposure to paramecia on the other hand triggered an even larger response in both LH lobes and led, as expected, to a significant reduction in cH activity. In order to quantify the relative changes in the mLH and lLH lobes, we compared the artemia-induced activity change ($\theta_A$) to the paramecia-induced activity change ($\theta_P$) for each lobe. The average mLH anti-pERK fluorescence only displayed a marginally greater artemia-induced increase ($\theta_A/\theta_P$ = 41%) than the lLH region ($\theta_A/\theta_P$ = 38%; *Figure 4c*, top panel). However, when the frequency of active neurons was compared, the mLH displayed a much larger response ($\theta_A/\theta_P$ = 32%) to artemia than the lLH ($\theta_A/\theta_P$ = 15%) (*Figure 4c*, bottom panel). Taken together with our calcium imaging results (*Figure 3*), these observations indicate that while all three hypothalamic regions (cH, mLH and lLH) are modulated by prey sensory cues, they respond more strongly to prey ingestion. Among these regions, the mLH appears to be the most highly tuned to prey detection in the absence of prey ingestion (*Figure 4d*).

## Optogenetic cH activation suppresses lLH neural activity

The observed anti-correlated patterns of caudal and lateral hypothalamus neural activity in both our calcium imaging and pERK-based activity data suggest they might interact via mutual inhibition. For example, during food deprivation, rising cH activity (and the absence of food) could restrain LH activity, while a subsequent experience of prey detection and ingestion might trigger LH activity that inhibits cH activity. This reduction in cH activity may, in turn, relieve suppression of LH activity, a neural 'switch' that could drive voracious feeding behavior.

As an initial test of this hypothesis, we determined whether optogenetic excitation of cH neurons would be sufficient to inhibit LH neural activity. We used the *Tg(y333:Gal4)* line (*Marquart et al., 2015*) to drive expression of a red-shifted channelrhodopsin (*Tg(UAS:ReaChR-RFP)*) (*Dunn et al., 2016*; *Lin et al., 2013*) in cH neurons (see *Figure 5—figure supplement 1* regarding choice of *Tg (y333:Gal4)*). The *Tg(y333:Gal4)* line drives ReaChR expression in a large fraction of cH serotonergic neurons (57.4 ± 2.1%; *Figure 5—figure supplement 1*), as well as a smaller fraction of dopaminergic cells (23.9 ± 2.2%; up to 30% overlap observed, *Figure 5—figure supplement 2*). *Tg(HuC: GCaMP6s)* was co-expressed to monitor spontaneous LH neuron calcium activity.

These tethered transgenic fish were subjected to targeted laser (633 nm) illumination of the cH region to locally activate the ReaChR channel. We showed that ReaChR activation in the cH was sufficient to induce cH neural activity (*Figure 5a,c*). In contrast, ReaChR activation significantly reduced spontaneous lLH calcium spike activity within a 90 s period that followed laser illumination (*Figure 5b,d*), whereas no significant decrease was observed in mLH activity (*Figure 5b,d*). Illumination of a control preoptic area region, where *Tg(y333:Gal4)*-driven ReaChR is not expressed, did not affect lLH activity, though we did observe a small increase in mLH activity (*Figure 5e*). This effect might be visually induced or driven by light-sensitive opsins known to be expressed in the preoptic area (*Fernandes et al., 2012*). Since no such increase was observed when the cH itself was optogenetically activated, it is plausible that an inhibitory effect of cH stimulation on the mLH is masked by an opposing light response sensitivity. In sum, optogenetic stimulation of cH neural activity is sufficient to inhibit lLH neural activity, consistent with the notion that cH and LH regions interact to modulate the animal's motivational state in response to food deprivation and feeding.

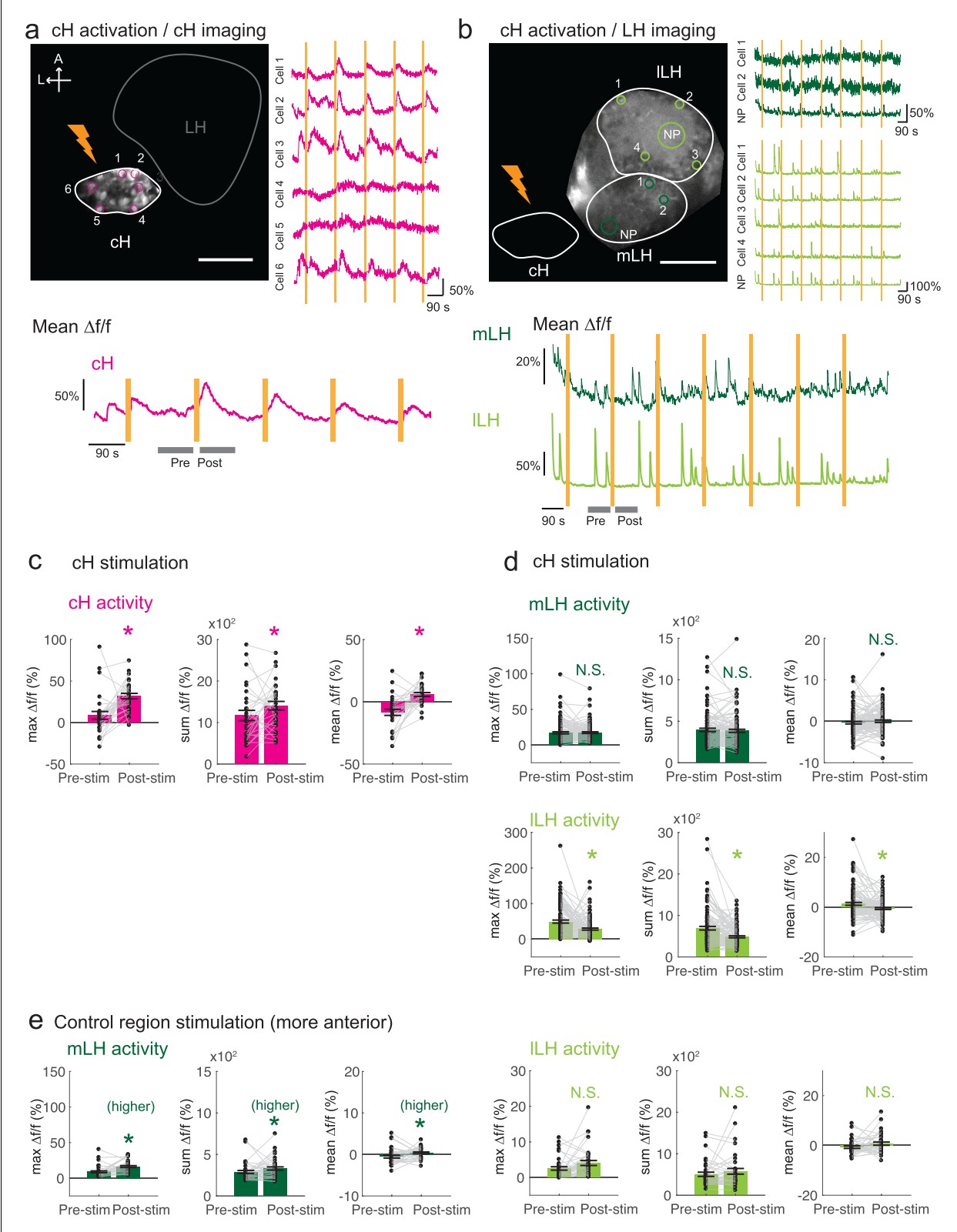

**Figure 4.** Sensory cues and prey ingestion differentially regulate cH and LH neural activity. (a) Representative images of activity induced by exposure of 7–8 dpf larval zebrafish to paramecia or artemia larvae, as examined by anti-pERK antibody staining. Hatched artemia are actively hunted but are too large to consume, allowing for the dissociation of sensory cues and hunting behavior from prey consumption. Scale bar = 50 μm. Rightmost two panels (top and bottom): Larval zebrafish hunt live artemia, performing J-turns and pursuits with eyes converged (see ***Video 4***; ***Bianco et al., 2011***). (b) cH

*Figure 4 continued on next page*

*Figure 4 continued*

activity (normalized pERK fluorescence intensity) is significantly reduced by exposure to paramecia but not by exposure to artemia (p = 0.016 (paramecia), 0.648 (artemia)). Asterisks denote p<0.05. (c) LH activity can be induced by artemia, and more strongly by paramecia. Both normalized pERK intensity (mLH: p = $2.06\times10^{-5}$ (paramecia vs control), p = $7.09\times10^{-4}$ (artemia vs control), p = $5.43\times10^{-5}$ (artemia vs paramecia); lLH: p = $2.06\times10^{-5}$ (paramecia vs control), p = 0.020 (artemia vs control), p = 0.0019 (artemia vs paramecia)) and active cell count (mLH: p = $2.06\times10^{-5}$ (paramecia vs control), p = $9.58\times10^{-5}$ (artemia vs control), p = $1.77\times10^{-4}$ (artemia vs paramecia); lLH: p = $2.06\times10^{-5}$ (paramecia vs control), p = $9.75\times10^{-5}$ (artemia vs control), p = $9.86\times10^{-5}$ (artemia vs paramecia)) are shown, with n = 9/9/11 fish, one-tailed Wilcoxon rank-sum test. Anti-pERK staining fluorescence was averaged over each entire region of interest (cH, mLH and lLH; see Materials and methods for details). The normalized anti-pERK staining intensity for each region (ROI) was obtained by dividing the anti-pERK fluorescence from each fish (in all experimental groups) by the average anti-pERK fluorescence for the same ROI of food-deprived (i.e. control) fish. We also compared the artemia-induced activity change ($\theta_A$) to the paramecia-induced activity change ($\theta_P$) for each lobe (see main text). (d) Differential neural activation of the cH and LH regions in response to prey sensation and hunting as compared to prey ingestion. Data plotted in *Figure 4* are provided in *Figure 4—source data 1*.
The online version of this article includes the following source data for figure 4:

**Source data 1.** Source data for plots displayed in *Figure 4b-c*.

## Functional dissection of the role of cH serotonergic neurons in feeding behavior

The opposing patterns of cH and LH activity suggest they might encode opposing functions in the motivation and control of feeding behavior. Increased cH activity during food deprivation might encode a motivated state that leads to enhanced prey detection, enhanced hunting behavior and increased prey ingestion following food presentation. In contrast, the incremental increase in cH activity during feeding (*Figure 1g*) might progressively inhibit lLH activity (*Figure 5*) and thus inhibit prey ingestion (*Muto et al., 2017*). To test these expectations, we used optogenetic ReaChR activation to increase cH neuron activity during food deprivation or during voracious feeding. We reasoned that since after a short period of food deprivation (≤30 minutes), cH activity is relatively low (*Figure 2a,b*), optogenetic cH neuron activation in such animals would mimic a longer food deprivation and yield subsequent voracious feeding. In contrast, animals that are already feeding voraciously will have very low cH activity (*Figures 1f–g* and *2a–b*); cH activation in these animals might thus reduce voracious feeding by mimicking the 'satiated' state (*Figure 1f,g*).

Accordingly, animals expressing ReaChR in cH neurons (*Tg(y333:Gal4;UAS:ReaChR-RFP)*) were exposed to 630 nm illumination and assessed for ingestion of fluorescently labeled paramecia (*Figure 6*). Such animals exhibited enhanced cH activity following illumination (*Figure 6*; *Figure 6—figure supplement 1*). As expected, animals that had been illuminated during a short period of food deprivation subsequently consumed significantly more paramecia than control fish, which were similarly food-deprived and illuminated, but lacked the ReaChR transgene (*Figure 6a*). In contrast, fish that had been illuminated at the end of a two-hour food deprivation period displayed a high level of prey ingestion irrespective of whether the ReaChR channel was present. Thus, the high level of cH activity produced by two hours long food deprivation could not be augmented by optogenetic activation.

On the other hand, when cH activity was optogenetically excited during voracious feeding (where cH activity would normally be very low), prey ingestion was reduced (*Figure 6b*). We presume that increased cH activity inhibits lLH activity (*Figure 5*), which in turn is associated with satiation and lack of feeding (*Figure 1f,g*). Indeed, inhibition of LH signaling has been shown to reduce prey capture success in comparable studies (*Muto et al., 2017*).

Finally, we asked what would happen if cH activity was reduced by partial ablation of serotonergic cells. Chemical-genetic ablation was performed via expression of a transgenic bacterial nitroreductase (*Tg(UAS:nfsb-mCherry)*) (*Curado et al., 2008*; *Davison et al., 2007*;

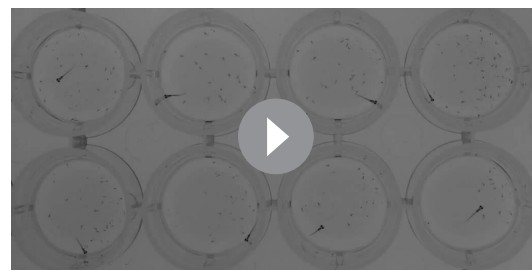

**Video 4.** Video of larval zebrafish hunting artemia larvae. Prey-capture behavior, such as J-turns and pursuits, but no capture swims, were observed in response to artemia larvae. Recording rate: 30 fps. Playback rate: Real time.
https://elifesciences.org/articles/43775#video4

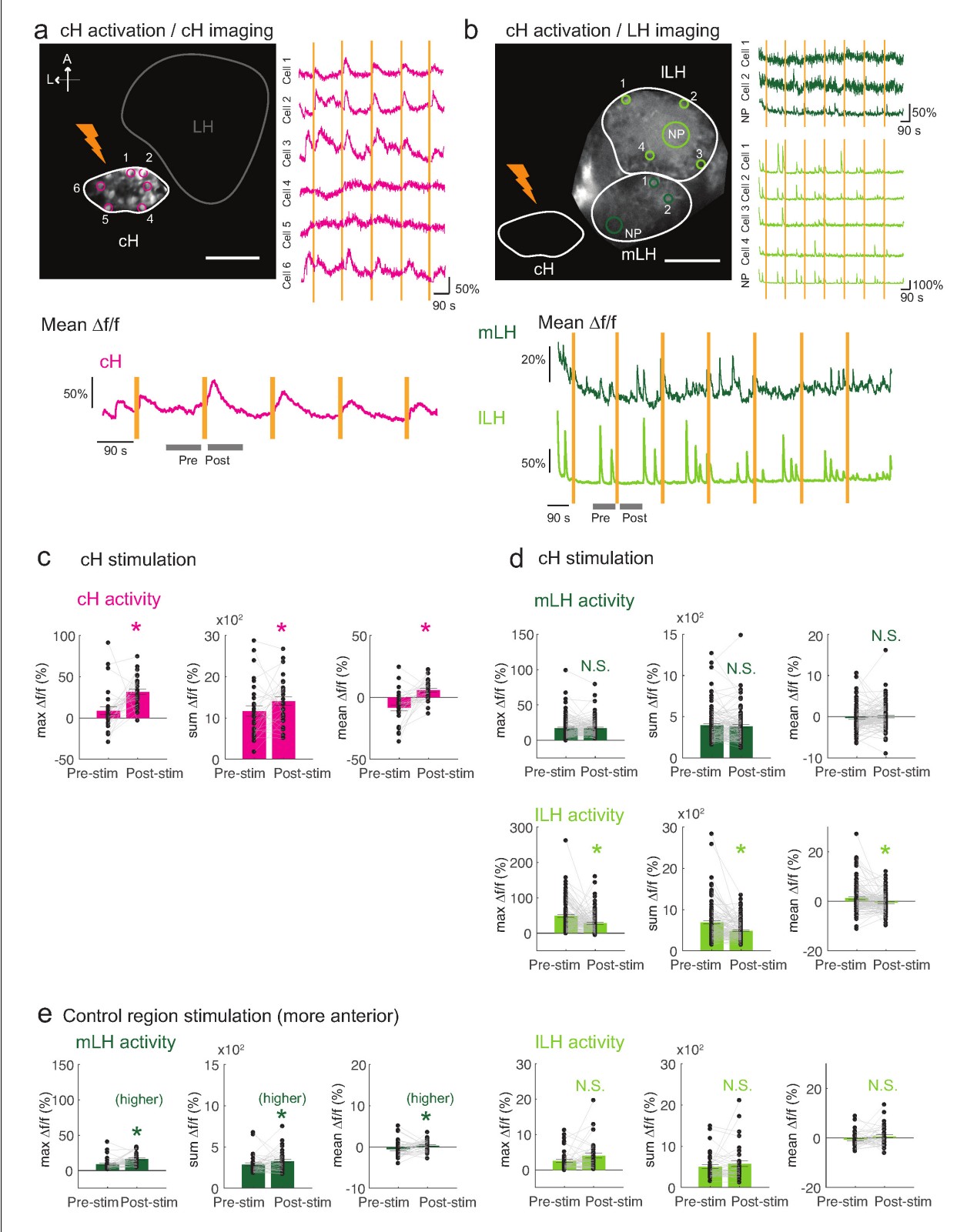

**Figure 5.** Optogenetic cH stimulation reduces lLH activity in tethered fish. (a) ReaChR activation of neurons. Top Panels: Targeted 633 nm laser illumination (see Materials and methods) of a defined cH area (imaged area) in *Tg(y333:Gal4;UAS:ReaChR-RFP; UAS:GCaMP6s)* fish. These animals express a *Tg(UAS:GCaMP6s)* reporter in the cH under *Tg(y333:Gal4)* control. The animals were subjected to repetitive 10 s laser illumination, with a periodicity of 120 s. Following the 633 nm laser pulses, there is widespread induction of cH activity, as indicated by GCaMP fluorescence (Δf/f) in most

*Figure 5 continued on next page*

*Figure 5 continued*

regions of interest plotted to the right of the image panel. Scale bar = 50 µm. Bottom Panel: Mean Δf/f across the entire outlined cH region versus time. Laser illumination pulses are indicated by orange bars. Gray bars indicate pre- and post-stimulation periods for which metrics shown in (c–e) were determined. (b) Inhibition of LH activity by activation of cH neurons in *Tg(y333:Gal4;UAS:ReaChR-RFP; HuC:GCaMP6s)* fish. The animals were subjected to repetitive 10 s laser illumination, with a periodicity of 180 s. Laser pulses were delivered to the cH (orange lightning symbol) as in a, and calcium imaging was recorded from the indicated LH areas (white outlines). Region of interest traces are shown to the right of the image panel for the indicated areas (**cells** and neuropil (**NP**)). There is an apparent reduction of spontaneous lLH GCaMP fluorescence spikes in the post-stimulation period. Scale bar = 50 µm. Bottom: Mean Δf/f across mLH and lLH ROIs over time. (c–e) Comparison of mean, summed and maximum Δf/f metrics for a 90 s window before and after ReaChR stimulation (gray bars in bottom panels in a and b). Each data point represents a single stimulation event, like those shown in a and b. Asterisks denote p<0.05. (c) cH activity increases after illumination of *Tg(y333:Gal4; UAS:ReaChR-RFP)*-positive cH neurons, n = 29 stimulations across eight fish, p = 0.0002 (max Δf/f) / 0.036 (sum Δf/f) / $9.2 \times 10^{-5}$ (mean Δf/f), one-tailed Wilcoxon signed-rank test. (d) lLH activity is inhibited (p = 0.0003 (max Δf/f) / $1.8 \times 10^{-6}$ (sum Δf/f) / 0.049 (mean Δf/f)), whereas mLH activity appears unchanged after after illumination of *Tg(y333:Gal4; UAS:ReaChR-RFP)*-positive cH neurons (p = 0.74 (max Δf/f) / 0.85 (sum Δf/f) / 0.13 (mean Δf/f)), n = 108 stimulations across nine fish, two-tailed Wilcoxon signed-rank test. (e) Illumination of a control preoptic region (outside of the area labeled by *Tg(y333:Gal4; ReaChR-RFP)* expression) resulted in a small increase in mLH activity (p = 0.0003 (max Δf/f) / 0.039 (sum Δf/f) / 0.039 (mean Δf/f)) and no change lLH activity (p = 0.099 (max Δf/f) / 0.65 (sum Δf/f) / 0.096 (mean Δf/f)), n = 37 stimulations across five fish, two-tailed Wilcoxon signed-rank test. Data plotted in *Figure 5* are provided in *Figure 5—source data 1*.

The online version of this article includes the following source data and figure supplement(s) for figure 5:

**Source data 1.** Source data for plots displayed in *Figure 5c-e*.
**Figure supplement 1.** Characterization of the serotonergic identity of the y333:Gal4 line.
**Figure supplement 2.** Characterization of the dopaminergic identity of the y333:Gal4 line.

*Pisharath and Parsons, 2009*) that was driven in cH serotonergic neurons by *Tg(116A:Gal4)* (*Figure 3—figure supplement 1*). *Tg(116A:Gal4; UAS:nfsb-mCherry)*-positive animals displayed a loss of nfsb-mCherry-expressing neurons after treatment with the chemical MTZ (*Figure 6—figure supplement 2*). These animals were compared to MTZ-treated sibling control animals lacking the *Tg(UAS: nfsb-mCherry)* transgene (*Figure 6c*). Fish with ablated cH serotonergic neurons displayed greater food ingestion than control animals irrespective of whether the animals had been food-deprived or continuously fed (*Figure 6c*). Animals that had been continuously fed displayed greater prey ingestion. They thus appear to display a defect in cH-mediated inhibition of feeding (*Figure 6b*) that could underlie satiety. Animals that had been food-deprived displayed greater than normal (relative to non-ablated control animals) voracious feeding (*Figure 6c*). Taken together, these results are consistent with the notion that cH activity regulates hunting and prey ingestion, at least partially via inhibition of hunting and prey ingestion behaviors.

## Discussion

Decades-old studies on appetite regulation in mammals have suggested that the hypothalamus consists of modular units that functionally interact to suppress or enhance food intake. Here we show that the larval zebrafish hypothalamic network can similarly be divided into medial and lateral units on the basis of neural activity and function. These units show anti-correlated activity patterns extending through various states and distinct behaviors during periods of food deprivation and feeding. We propose these states are analogous to those commonly referred to as hunger and satiety and reflect the animal's drive to maintain energy homeostasis (*Figure 6d*). Furthermore, we show that within these broad neural response classes lie subpopulations that encode specific stimuli and perform distinct functions depending on the timing of their activation.

### Mutually opposing hypothalamic networks control zebrafish appetite

We show that the medial hypothalamic zone, especially the caudal hypothalamus (cH), is strongly activated by food deprivation and silent during voracious feeding, and that these changes in activity occur on a timescale of seconds to minutes. Here, we focused mainly on the cH serotonergic neurons, although many medially localized neurons show similar activity patterns. In contrast, the lateral hypothalamus (LH), which contains GABAergic and glutamatergic neurons, can be inhibited by the cH (*Figure 5*) and is weakly active in the absence of food; conversely it is most strongly active during voracious feeding when cH serotonergic neurons are silent. Interestingly, fish that display satiated feeding behavior exhibit intermediate activity levels in the two hypothalamic regions (*Figure 1*). Thus, "hunger" in the larval zebrafish is encoded by two alternative and distinct states of activity in

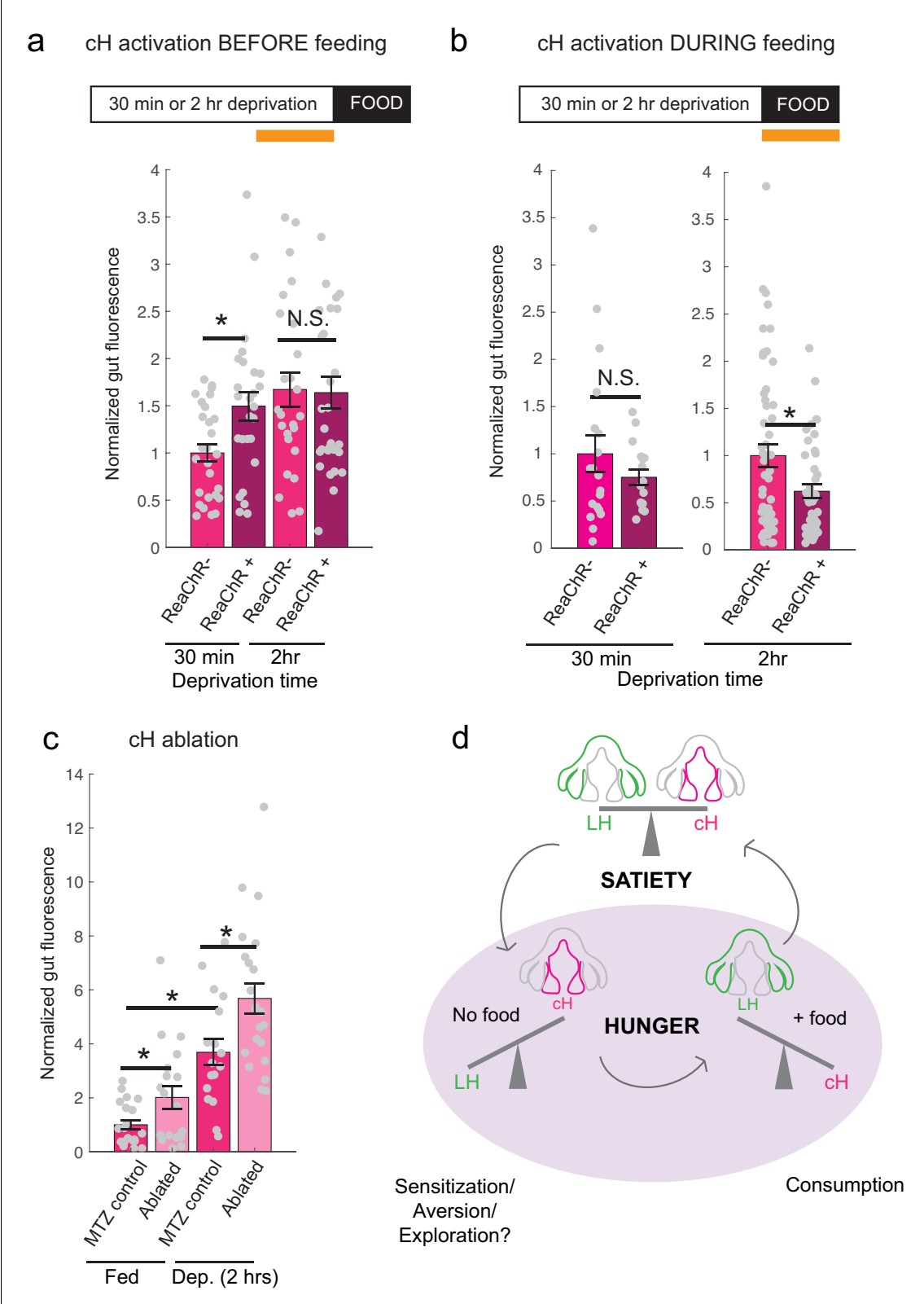

**Figure 6.** Role of the cH in behavioral control. (**a**) Animals expressing the ReaChR transgene *Tg(UAS:ReaChR-RFP)* under control of the *Tg(y333:Gal4)* driver were exposed to 630 nm illumination (orange bar in schematic) for 10 min prior to feeding and assessed for subsequent ingestion of fluorescently labeled paramecia. *Tg(y333:Gal4; UAS:ReaChR-RFP)* stimulation increased food intake in 30 min food-deprived but not 2 hr food-deprived fish, during subsequent food presentation. Dep. (30 min): n = 27/26 (ReaChR-/ReaChR+), p = 0.005. Dep. (2 hr): n = 25/29 (ReaChR-/ReaChR+), p = 0.36, one-tailed

*Figure 6 continued on next page*

*Figure 6 continued*

Wilcoxon rank-sum test. Asterisks denote p<0.05. Since ReaChR expression via *Tg(116A:Gal4)* was negligible, we used another Gal4 (*Tg(y333:Gal4)*) line that is also specific to the cH when ReaChR is expressed. Fed and food-deprived fish were assayed simultaneously, thus all results were normalized to fed controls. ReaChR- controls do not have visible *Tg(y333:Gal4;UAS:ReaChR-RFP)* expression, and thus are a mixture of siblings expressing *Tg(y333: Gal4) only*, *Tg(UAS:ReaChR-RFP)* or neither of these transgenes, each with ⅓ probability. (b) Left: Optogenetic activation of *Tg(y333:Gal4; UAS:ReaChR-RFP)* fish (orange bar in schematic) during feeding in fish that were food-deprived for 30 min does not significantly reduce food intake: n = 19/16 (ReaChR-/ReaChR+), p = 0.44 (N.S.); Right: Optogenetic activation of *Tg(y333:Gal4; UAS:ReaChR-RFP)* fish during feeding in 2 hr food-deprived fish reduces food intake: n = 53/44 (ReaChR-/ReaChR+), p = 0.042. Since 30 min and 2 hr food-deprived fish were assayed in different experiments, gut fluorescence normalized to their respective controls, one-tailed Wilcoxon rank-sum test. (c) Nitroreductase-mediated ablation of the cH in *Tg(116A: Gal4;UAS:nfsb-mCherry)*-positive or negative fish treated with metronidazole (MTZ) from 5 to 7 dpf significantly enhances food intake in 8 dpf fish. p = 0.0042/0.041/1.4 × 10$^{-5}$ (fed control vs fed ablated, 2 hr dep. control vs 2 hr dep. ablated, fed vs 2 hr dep.), n = 29 (fed control)/28 (fed ablated)/22 (dep. control)/29 (dep. ablated), two-tailed Wilcoxon rank-sum test. Controls do not have visible *Tg(116A:Gal4;UAS:nfsb-mCherry)* expression, and thus are a mixture of siblings expressing *Tg(116A:Gal4) only*, *Tg(UAS:nfsb-mCherry)* or neither of these transgenes, each with ⅓ probability. (d) Schematic summarizing our results. We propose distinct roles of the cH during hunger, depending on the presence or absence of food. See Appendix 1 – Conceptual Circuit Model for elaboration. Data plotted in *Figure 6* are provided in *Figure 6—source data 1*.

The online version of this article includes the following source data and figure supplement(s) for figure 6:

**Source data 1.** Source data for plots displayed in *Figure 6a-c*.
**Figure supplement 1.** ReaChR activation by whole-field optogenetic illumination.
**Figure supplement 2.** Nitroreductase-mediated ablation of cH serotonergic neurons.

opposing brain regions, depending on whether food is absent or present, with the restoration of energy homeostasis (i.e. satiety) paralleled by a return to an intermediate state of balanced activity.

While generally anti-correlated, the cH and LH also appear to be differentially modulated both by internal energy states and external factors such as prey. In the absence of food, LH neural activity decreases rapidly (*Figure 2*), suggesting a requirement of external food cues to drive LH activity, though some modest rate of spontaneous activity is still observed (*Figure 5*, *Figure 3—figure supplement 3*). In contrast, the slower timescale of increasing cH activity during food deprivation (*Figure 2*, *Figure 3—figure supplement 3*) may reflect a rising caloric deficit. Notably, many of the cH neurons are cerebrospinal fluid-contacting and thus have access to circulatory nutrient and hormone information (*Lillesaar, 2011*; *Pérez et al., 2013*).

When prey is presented to a food-deprived animal, a rapid state change occurs as LH neural activity is strongly increased and cH activity rapidly diminishes (*Figures 1–4*). Importantly, the silence of cH neurons and strength of LH activity were correlated with the extent of prior food deprivation (*Figure 2*), suggesting a role for these nuclei in regulating food intake based on energy needs. The quick timescale of these changes in activity suggests that they do not reflect an alleviation of caloric deficit (i.e. a change in hunger state), which would take a significantly longer time to occur. Further, the striking anti-correlation between the cH and LH is consistent with their mutual inhibition, and suggests that the acute reduction in cH activity allows for rapid LH excitation upon the presentation of prey cues. We supported this notion by showing that optogenetic stimulation of a subset of cH neurons could inhibit lLH activity (*Figure 5*). However, the mechanisms for cH and LH mutual interactions are still unknown. It is possible that the cH may act via nearby inhibitory GABAergic neurons, and/or exert its effects through direct secretion of monoamines into the ventricles or perineuronal space. The fast (seconds) anti-correlation between cH and LH calcium activity (*Figure 3*), suggests the presence of direct inhibitory connections. The LH, which was previously characterized in *Muto et al. (2017)*, similarly does not appear to send direct projections to the cH, but could potentially interact via intermediary neurons in the medial/periventricular regions of the hypothalamus.

## The cH and LH show differential sensitivity to prey sensory and consummatory cues

Ingestive behavior has been proposed to comprise a series of sequential phases: 1) the initiation phase, triggered by energy deficit, in which the animal begins to forage; 2) the procurement phase, triggered by the presence of food sensory cues, in which the animal seeks and pursues food; and 3) the consummatory phase, which usually involves more stereotyped motor programs (*Berthoud, 2002*; *Watts, 2000*). An animal's energy status is sensed internally and may influence the

initiation, procurement and consummatory stages of ingestive behavior. Thus, a hungry animal will be more alert to food cues, seek food more persistently and also eat more voraciously.

In mammals, LH neurons are responsive to both external food sensory cues and consummatory cues (*Jennings et al., 2015*). Here, we show that the LH lobes in zebrafish also respond to both types of food cues. In the 'sensory' stage, the mLH and lLH are already activated, which may reflect an enhanced sensitivity to food cues during hunger. In contrast, cH activity transiently falls (as shown by calcium imaging in *Figure 3*) but remains overall high.

Notably, cH inhibition and LH activation during the sensory stage is not as strong as post-food consumption (*Figure 4*), which induces massive and opposing changes in the activity of both domains. Since LH and cH activity are modulated within minutes of food consumption, they are unlikely to reflect satiety signals, and rather might play a role in further driving voracious food consumption, at least until the activity of both populations returns to baseline. While it is unclear which consummatory cues modulate LH and cH activity, based on live imaging results from *Muto et al. (2017)*, the greatest enhancement of LH activity was observed almost immediately (milliseconds to seconds) after paramecia consumption. Thus, the cue is likely a fast pregastric signal (taste/tactile/ swallowing), rather than postgastric absorption or hormone secretion.

Finally, our data raise the possibility of functional compartmentalization within the LH. Especially in terms of cellular pERK activity, the lLH is more weakly activated by food sensory cues compared to the mLH, suggesting that the lLH, similar to the cH, may be more sensitive to consummatory cues than sensory food cues alone. These results are also consistent with a generally stronger anti-correlation of lLH and cH activity (compared to mLH), as observed in our calcium imaging and optogenetic experiments. Further molecular, cellular, and functional dissection of the individual LH lobes will allow for a better understanding of their behavioral roles.

## Functional roles of the cH and LH in and beyond appetite control

Finally, we test the hypothesis that the cH and LH form mutually antagonistic functional units that dominate different phases of hunger and drive appropriate behavioral responses during each phase (*Figure 6*). In particular, we show that the activation state of the cH is a crucial regulator of satiation state-dependent food intake. Artificial cH activation in satiated fish *prior* to feeding is sufficient to drive subsequent voracious feeding. Based on observed cH dynamics, we propose that the degree of cH inhibition during voracious feeding is proportional to the degree of cH activation prior to feeding. This could be mediated by the release of serotonin/other neuromodulators over the course of food deprivation, which may be capable of sensitizing the LH even in the absence of food cues. In this way, zebrafish are able to retain a 'memory' of their hunger state, which is released once food is presented. This motif might help ensure that the animal eventually returns to a stable equilibrium, that is, satiety.

We furthermore show that the acute effect of cH activation *during* feeding is suppression of food intake, whereas cH ablation enhances food intake, which is again consistent with mammalian studies of medial hypothalamic areas. At first glance, the observation that the cH acutely suppresses food intake is inconsistent with the idea that it is most active during food deprivation. However, the critical difference here is the presence or absence of food. Once food is presented to a hungry fish, high activity in the cH may simply suppress LH activity, and hence elevate the initial threshold for food intake.

The seemingly paradoxical roles of the cH during hunger may also make sense when considering that, in the absence of food, consummatory behavior would in fact be counterproductive. Thus, during food deprivation, the cH may play complementary roles such as the sensitization of the LH and/ or other feeding-related circuits (as discussed above), or drive alternative behavioral programs, like foraging or energy-conserving measures (see Appendix 1 - Conceptual Circuit Model for a more in-depth discussion). Given that cH neurons are also activated by aversive stimuli (*Randlett et al., 2015*; *Wee et al., 2019*), they might generally encode a negative valence state, of which being hungry in the absence of food is an example. The silence of these neurons in a hungry fish where food is present may then imply a positive valence state, a notion that is in ready agreement with human subjective experience. Similar features of hunger-related (i.e. AgRP) neurons have also been described in mammals (*Betley et al., 2015*; *Chen et al., 2015*; *Dietrich et al., 2015*; *Mandelblat-Cerf et al., 2015*).

Although the cH does not have an exact mammalian homolog, its functions have been proposed to be adopted by other modulatory populations, such as the serotonergic raphe nucleus in mammals (*Gaspar and Lillesaar, 2012*; *Lillesaar, 2011*). While shown to be a potent appetite suppressant, serotonin is also released during food deprivation, and can enhance food-seeking behavior (*Elipot et al., 2013*; *Kantak et al., 1978*; *Pollock and Rowland, 1981*; *Voigt and Fink, 2015*). Thus, our results revealing opposing cH activity patterns during hunger could reflect similarly complex roles of serotonin in zebrafish, potentially explaining some of its paradoxical effects on food intake and weight control in mammals (*Harvey and Bouwer, 2000*). The cH and PVO also express dopaminergic (intermingled with 5-HT) and a much smaller fraction of histaminergic neurons, which appear to be densely interconnected (*Chen et al., 2016*; *Kaslin and Panula, 2001*). We note that our data, while confirming a role of serotonergic neurons, does not rule out an involvement of these other neuromodulators in appetite control, particularly dopamine.

Further, we do not rule out the involvement of other circuits in appetite control; in fact, there are likely numerous players involved. For example, the PVO appears to be modulated by food cues and food deprivation, is anti-correlated with LH activity, and labeled by our transgenic lines (albeit more sparsely), suggesting it may complement the role of the cH. Our conclusions are also limited by the available tools and methodologies – since different transgenic lines were utilized for stimulation and ablation, we cannot be certain that we are manipulating the same population of neurons, though both share mutual overlap with serotonergic cells. Also, due to the lack of complete transgene specificity, there is a possibility that our manipulations may affect non-specific targets such as the olfactory bulb.

The strong LH activation by the presentation of food after food deprivation suggests that this region is involved in the induction of voracious feeding. This notion is supported by *Muto et al. (2017)* who recently demonstrated that inhibition of the LH impairs prey capture, a behavior that is clearly related to voracious feeding. Furthermore, electrical stimulation of the homologous region (lateral recess nuclei) in adult cichlids and bluegills (*Demski, 1973*; *Demski and Knigge, 1971*) can elicit feeding behavior, which is consistent with our hypothesis. Interestingly, while stimulating parts of this region induced food intake, the activation of other parts induced behaviors such as the 'snapping of gravel', which are reminiscent of food search or procurement. In mammals, electrical or optogenetic stimulation of LH neurons triggers voracious feeding, again consistent with our findings that the LH is highly activated during the voracious feeding phase in hungry fish (*Delgado and Anand, 1952*). In particular, GABAergic neurons that do not co-express MCH or Orexin have been shown to be responsive to food cues and are sufficient to stimulate food intake in mammals (*Jennings et al., 2015*). Whether the GABAergic and glutamatergic neurons of the zebrafish LH co-express other neuromodulators, as has been recently discovered in mammals (*Mickelsen et al., 2019*) remains to be explored. Overall, these data suggest that the zebrafish LH may play an important role in driving food intake during hunger, despite some differences in peptidergic expression from the mammalian LH. Certainly, since cues such as water flow and optogenetic stimulation light are sufficient to modulate cH and/or LH neurons, these hypothalamic loci may be also involved in other sensorimotor behaviors beyond appetite regulation.

In conclusion, we have shown here how anatomically-segregated hypothalamic nuclei might interact to control energy homeostasis. We argue that the medial-lateral logic of hypothalamic function that is well established in mammalian systems may be conserved even in non-mammalian vertebrates, though their activity patterns might possibly be more complex than originally believed. Our data suggest diverse roles of neuromodulators such as serotonin in regulating behavioral responses during hunger, which complement mammalian observations. Finally, we propose that investigating large-scale network dynamics can reveal an additional layer of insight into the principles underlying homeostatic behavior, which might be overlooked when studies are restricted to the observation and perturbation of smaller subpopulations.

## Materials and methods

**Key resources table**

*Continued*

| Reagent type (species) or resource | Designation | Source or reference | Identifiers | Additional information |
|---|---|---|---|---|
| Reagent type (species) or resource | Designation | Source or reference | Identifiers | Additional information |
| Genetic reagent (*Danio rerio*) | Tg(pGal4FF:116A) | Characterized in this manuscript | | Dr. Koichi Kawakami (NIG, Japan) |
| Genetic reagent (*Danio rerio*) | Tg(pGal4FF:76A) | PMID: 28425439 | | Dr. Koichi Kawakami (NIG, Japan) |
| Genetic reagent (*Danio rerio*) | Tg(y333:Gal4) | PMID: 26635538 | | Dr. Harold Burgess (NIH) |
| Genetic reagent (*Danio rerio*) | Tg(HuC:GCaMP6s) | PMID: 28892088 | | Dr. Florian Engert (Harvard) |
| Genetic reagent (*Danio rerio*) | Tg(UAS:GCaMP6s) | PMID: 28425439 | | Dr. Koichi Kawakami (NIG, Japan) |
| Genetic reagent (*Danio rerio*) | Tg(UAS:ReaChR-RFP) | Characterized in this manuscript | | Dr. Misha Ahrens (Janelia Research Campus) |
| Genetic reagent (*Danio rerio*) | Tg(UAS-E1b:NTR-mCherry) | PMID: 17335798 | | Available from ZIRC |
| Genetic reagent (*Danio rerio*) | Tg(Vglut2a:dsRed) | PMID: 19369545 | | |
| Genetic reagent (*Danio rerio*) | Tg(Gad1b:loxP-dsRed-loxP-GFP) | PMID: 23946442 | | |
| Genetic reagent (*Danio rerio*) | Tg(Gad1b:GFP) | PMID: 23946442 | | |
| Genetic reagent (*Danio rerio*) | Tg(TH2:GCaMP5) | PMID: 26774784 | | Dr. Adam Douglass (University of Utah) |
| Genetic reagent (*Danio rerio*) | Tg(ETvmat2:GFP) | PMID:18164283 | | |
| Genetic reagent (*Danio rerio*) | Tg(HCRT:RFP) | PMID: 25725064 | | |
| Antibody | rabbit monoclonal anti-pERK | Cell Signaling | 4370 RRID:AB_2315112 | IHC (1:500) |
| Antibody | mouse monoclonal anti-ERK | Cell Signaling | 4696 RRID:AB_390780 | IHC (1:500) |
| Antibody | rabbit polyclonal anti-5-HT | Sigma-Aldrich | S5545 RRID:AB_477522 | IHC (1:500) |
| Antibody | goat polyclonal anti-5-HT | AbCam | ab66047 RRID:AB_1142794 | IHC (1:500), 2% BSA in PBS, 0.3% Triton blocking solution) |
| Antibody | goat polyclonal anti-MSH | EMD Millipore | AB5087 RRID:AB_91683 | IHC (1:500), 2% BSA in PBS, 0.3% Triton blocking solution) |
| Antibody | rabbit polyclonal anti-AGRP | Phoenix Pharmaceuticals | H-003–53 RRID:AB_2313908 | IHC (1:500) |
| Antibody | rabbit polyclonal anti-MCH | Phoenix Pharmaceuticals | H-070–47 RRID:AB_10013632 | IHC (1:500) |
| Antibody | rabbit polyclonal anti-CART | Phoenix Pharmaceuticals | 55–102 RRID:AB_2313614 | IHC (1:500) |
| Antibody | rabbit polyclonal anti-NPY | Immunostar | 22940 RRID:AB_2307354 | IHC (1:500) |
| Antibody | mouse monoclonal anti-TH | Immunostar | 22941 RRID:AB_1624244 | IHC (1:500) |

*Continued on next page*

*Continued*

| Reagent type (species) or resource | Designation | Source or reference | Identifiers | Additional information |
|---|---|---|---|---|
| Chemical compound, drug | DiD' solid (lipid dye) | Thermo Fisher Scientific | D-7757 | Stock solution (10 mg/ml), working solution (2.5 mg/ml), in ethanol |

## Fish husbandry and transgenic lines

Larvae and adults were raised in facility water and maintained on a 14:10 hr light:dark cycle at 28°C. All protocols and procedures involving zebrafish were approved by the Harvard University/Faculty of Arts and Sciences Standing Committee on the Use of Animals in Research and Teaching (IACUC). WIK wildtype larvae and mit1fa-/- (nacre) larvae in the AB background, raised at a density of ~40 fish per 10 cm petri dish, were used for behavioral and MAP-mapping experiments.

Transgenic lines *Tg(UAS-E1b:NTR-mCherry)* (*Davison et al., 2007*) (referred to as UAS:nfsb-mCherry), *Tg(UAS:GCaMP6s)* (*Muto and Kawakami, 2011*; *Muto et al., 2017*) *Tg(HuC:GCaMP6s)* (*Kim et al., 2017*), *Tg(Vglut2a:dsRed)* (*Miyasaka et al., 2009*), *Tg(Gad1b:loxP-dsRed-loxP-GFP* and *Tg(Gad1b:GFP)* (*Satou et al., 2013*), *Tg(TH2:GCaMP5)* (*McPherson et al., 2016*), *Tg(ETvmat2:GFP)* (referred to as VMAT:GFP) (*Wen et al., 2008*), *Tg(HCRT:RFP)* (*Liu et al., 2015*) have all been previously described and characterized. *Tg(pGal4FF:116A)* (referred to as 116A:Gal4) was isolated from a gene trap screen by the Kawakami group (*Kawakami et al., 2010*), *Tg(pGal4FF:76A)* was recently published by the same group (*Muto et al., 2017*). *Tg(y333:Gal4)* from a different enhancer trap screen was used to drive expression in the cH in cases where 116A:Gal4-driven expression was sparse (*Marquart et al., 2015*). *Tg(UAS:ReaChR-RFP)* was generated by Chao-Tsung Yang (Ahrens lab, Janelia Research Campus) using Tol2 transgenesis. The same optogenetic channel was previously validated in zebrafish in *Dunn et al. (2016)*.

## MAP-mapping of appetite regions

More details on the MAP-mapping procedure can be found in *Randlett et al. (2015)*. 5–6 dpf, *mit1fa-/-* (nacre) larvae in the AB background were fed an excess of paramecia once daily. On the day of the experiment (at 7 dpf), the larvae were distributed randomly into two treatment groups: 1) Food-deprived, where larvae were transferred into a clean petri dish of facility water, taking care to rinse out all remaining paramecia or 2) Fed, where after washing and transferring they were fed again with an excess of paramecia. After two hours, larvae in both groups were fed with paramecia. After 15 min, larvae were quickly funneled through a fine-mesh sieve, and the sieve was then immediately dropped into ice-cold 4% paraformaldehyde (PFA) in PBS (PH 7.2–7.4). Fish were then immunostained with procedures as reported below (see Immunostaining methods). The rabbit anti-pERK antibody (Cell Signaling, #4370) and mouse anti-ERK (p44/42 MAPK (Erk1/2) (L34F12) (Cell Signaling, #4696) were used at a 1:500 dilution. Secondary antibodies conjugated with alexa-fluorophores (Life Technologies) were diluted 1:500. For imaging, fish were mounted dorsal-up in 2% (w/v) low melting agarose in PBS (Invitrogen) and imaged at ~0.8/0.8/2 µm voxel size (x/y/z) using an upright confocal microscope (Olympus FV1000), using a 20 × 1.0 NA water dipping objective. All fish to be analyzed in a MAP-Mapping experiment were mounted together on a single imaging dish, and imaged in a single run, alternating between treatment groups.

## ICA analysis

ICA analysis was performed exactly as reported in *Randlett et al. (2015)*. The central brain (not including eyes, ganglia, or olfactory epithelia) from each fish was downsampled into 4.7 um$^3$ sized voxels to generate a pERK level vector for each fish. Fish in which any of the voxels was not imaged (due to incomplete coverage) were excluded from the analysis. Fish were normalized for overall brightness by dividing by the 10th percentile intensity value, and voxels normalized by subtracting the mean value across fish. The fish-by-voxel array was then analyzed for spatially independent components using FastICA (http://research.ics.aalto.fi/ica/fastica/, Version 2.5), treating each fish as a signal and each voxel as sample, using the symmetric approach, 'pow3' nonlinearity, retaining the

first 30 principal components and calculating 30 independent components. Independent component (IC) maps are displayed as the z-score values of the IC signals.

Since ICA analysis requires a substantial sample size, the original analysis reported in *Randlett et al. (2015)* included 820 fish exposed to various treatments, including fish sampled at different points of the day and night, and fish given various noxious or food stimuli, additional fish stimulated with electric shocks, light flashes, moving gratings, heat, mustard oil, melatonin, clonidine, nicotine, cocaine, ethanol and d-amphetamine.

Here, to focus the analysis on more naturalistic feeding conditions, we restricted the dataset to n = 300 fish that were either food-deprived (2 hr), or presented with food in food-deprived or fed conditions.

## Whole-mount immunostaining

24 hr after fixation (4% paraformaldehyde (PFA) in PBS), fish were washed in PBS + 0.25% Triton (PBT), incubated in 150 mM Tris-HCl at pH 9 for 15 min at 70˚C (antigen retrieval), washed in PBT, permeabilized in 0.05% Trypsin-EDTA for 45 min on ice, washed in PBT, blocked in blocking solution (10% Goat Serum, 0.3% Triton in Balanced Salt Solution or 2% BSA in PBS, 0.3% Triton) for at least an hour and then incubated in primary and secondary antibodies for up to 3 days at 4˚C diluted in blocking solution. In-between primary and secondary antibodies, fish were washed in PBT and blocked for an hour. If necessary, pigmented embryos were bleached for 5 min after fixation with a 5%KOH/3%$H_2O_2$ solution.

The protocol was similar for dissected brains, except that the brains were dissected in PBS after 24 hr of fixation, and the permeabilization step in Trypsin-EDTA and occasionally Tris-HCL antigen retrieval were omitted. Dissected brains were mounted ventral up on slides in 70% glycerol prior to imaging. Confocal images of dissected brains were obtained using either a Zeiss LSM 700 or Olympus FV1000.

## Quantification of food intake

Paramecia cultures (~1–2 500 ml bottles) were harvested, spun down gently (<3000 rpm) and concentrated, and subsequently incubated with lipid dye (DiD' solid, D-7757, Thermo Fisher Scientific, dissolved in ethanol) for >2 hr (5 µl of 2.5 mg/ml working solution per 1 ml of concentrated paramecia) on a rotator with mild agitation. They were then spun down gently (<3000 rpm), rinsed and reconstituted in deionized water. An equal amount (100 µl,~500 paramecia) was pipetted into each 10 cm dish of larvae. This method was adapted from *Shimada et al. (2012)*. After the experiment, larvae were fixed and mounted on their sides on glass slides or placed in wells of a 96 well plate. They were then imaged using the AxioZoom V16 (Zeiss) and analyzed using custom Fiji (*Schindelin et al., 2012*) software. In cases where the identity of larvae needed to be maintained, for example, to correlate food intake with brain activity, larvae were imaged and subsequently stained individually in 96 well plates. This led to more variable staining which affects analysis of mean fluorescence.

Larvae were always distributed randomly into experimental groups.

## Quantification of LH and cH activity in dissected brains

Brains within each dataset were usually registered onto a selected reference image from the same dataset using the same CMTK registration software used in MAP-mapping. Further analysis was then performed using custom Fiji and MATLAB software.

### Quantification of mean anti-pERK fluorescence

For quantification of cH, mLH and lLH pERK fluorescence intensity, ROIs were manually defined using the reference image, and pERK intensity was quantified over all registered images and averaged across the entire lobe (multiple z-planes) as well as across both lobes. Analysis of cH pERK fluorescence was restricted to the most ventral planes, as more dorsal cH neurons show weaker correlation with feeding states (e.g. *Figure 1—figure supplement 5*).

## Quantification of active cell count

For quantification of mLH and lLH active cell count, automated analysis of cell count was again performed using custom Fiji software, namely: 1) Image processing to reduce background and enhance contrast 2) Adaptive thresholding to isolate strongly-stained cells 3) Applying the 'Analyze Particles' function to quantify the number of cells within each manually-defined ROI. Aggregation and visualization of results were performed using custom MATLAB software.

Note that, in experiments in which the data were collected without the tERK channel (e.g. from *Figure 2*), thus prohibiting image registration, ROIs were drawn manually over each region across all z-planes and averaged to obtain mean fluorescence values. For *Figure 2—figure supplement 1*, where individual fish were stained, all measurements, including cell count, were made manually. In addition, background fluorescence was measured for each sample and subtracted from measured values.

## Semi-automated quantification of ReAChR overlap with anti-pERK staining

This section describes the analysis method for *Figure 6—figure supplement 1*. The multi-point picker on ImageJ was first used to select all visible ReAChR-positive or ReAChR-negative cells within each z-stack for each fish. A custom Fiji macro was then used to extract mean pERK intensities from all identified cells, and data were further processed using MATLAB. Data were plotted using the not-BoxPlot.m Matlab function.

### Calcium imaging

For confocal calcium imaging of the cH and LH simultaneously in the presence of food, *Tg(76A: Gal4;116A:Gal4; UAS:GCaMP6s)* triple transgenic fish were embedded in 1.8% agarose, with their eyes/nostrils released. GCaMP activity from a single z-plane (where the cH and LH neurons could be seen) was imaged using a confocal microscope (Olympus FV1000) at one fps. After a 5 min habituation period and a 5 min baseline period, a dense drop of water, followed by paramecia (5 min later) was pipetted into the dish. Due to paramecia phototaxis, most of the paramecia moved into close vicinity of the fish's head under the laser, allowing for strong visual/olfactory exposure to paramecia. After image registration (TurboReg Fiji Plugin, *Thévenaz et al., 1998*), and downsampling (Fiji/MATLAB), manually-segmented ROIs were selected and total fluorescence within the ROI was calculated. Cross-correlation and other analyses were performed using custom MATLAB software.

For long-term 2P imaging of the cH and LH simultaneously in the absence of food (*Figure 3—figure supplement 3*), *Tg(76A:Gal4;116A:Gal4; UAS:GCaMP6s)* triple transgenic fish were embedded in 1.8% agarose. GCaMP activity from either multiple slices (3 z-planes spanning a ~ 20 µm volume of the intermediate hypothalamus using an electrically-tunable liquid lens (Edmund Optics, 83–922), 237 ms per z-plane) or a single z-plane where the cH and LH neurons (1.5 fps) could be seen was imaged using custom 2P microscopes. After image registration (Fiji/MATLAB), manually segmented ROIs were selected and total fluorescence within the ROI was calculated. Calcium spike detection and other analyses were performed using custom MATLAB software. Baseline detrending was performed on 'raw' Δf/f traces by fitting a quadratic polynomial and subtracting it from the trace. Calculations on calcium spike frequency and amplitude were subsequently performed using baseline-detrended calcium traces.

### Optogenetic stimulation and simultaneous calcium imaging

Optogenetic stimulation and calcium imaging was performed on a confocal microscope (Zeiss LSM 880) using a 633 nm laser for ReAChR activation, and a 488 nm laser for calcium imaging. *Tg(y333: Gal4;UAS:ReAChR-RFP; HuCGCaMP6s)* triple-transgenic fish were used to record LH activity after ReAChR activation. As *Tg(HuC:GCaMP6)* does not label the cH, in some cases we used fish that also had *Tg(UAS:GCaMP6s)* co-expressed in the cH, allowing for monitoring of cH activity directly.

The ReAChR activation spectrum is wide and 488 nm laser power at sufficiently high intensities is sufficient to activate ReAChR. Since *Tg(y333:Gal4;UASGCaMP6s)* is expressed strongly in the cH, weak 488 nm laser power can be used to monitor cH activity after ReAChR activation of cH. On the other hand, *Tg(HuC:GCaMP6s)* expression in the LH is considerably weaker than *Tg(UAS:GCaMP6s)* expression driven by *Tg(y333:Gal4)*, and recording LH activity requires high laser power. Thus, during LH recording trials, we could not simultaneously image the cH.

Fed fish were embedded in 1.8–2% agarose, with tails, mouth and eyes freed, 15–20 min before imaging in the absence of food. For baseline recording, spontaneous activities in cH or LH were recorded. ReachR activation was then induced in one side of cH periodically for 10–15 s, and ensuing activity in one or both sides of LH or cH was recorded continuously during intervals (of 120–180 s) between stimuli.

### Nitroreductase-mediated ablations

Larvae expressing *Tg(116A:Gal4;UAS:nfsb-mCherry)*, or their non-transgenic siblings were incubated in 2.5 mM Metronidazole (Sigma-Aldrich, M3761) from 4-6 dpf/5–7 dpf. MTZ was subsequently washed out, and food intake was measured at 7 or 8 dpf. For these experiments, the MTZ-treated non-transgenic siblings were used as the control group. Each control or ablated group was food-deprived or fed for 2 hr, and labeled food was added to quantify food intake. In the case of fed fish, unlabeled food was very gently washed out 15 mins before the experiment and the food-deprived fish were also agitated slightly to simulate a short washout.

### Optogenetic stimulation with behavior

Optogenetic stimulation was done by placing a square LED panel (630 nm, 0.12 mW/mm$^2$ driven at full current, Soda Vision, Singapore) directly on top of petri dishes containing ReAChR positive or negative fish, for 10 min continuously before or during feeding. We had attempted other methods of stimulating the fish (e.g. pulsed LED stimulation) but found that it was disruptive to behavior.

### Artemia hunting video

7 dpf larval fish were food-deprived for 2 hr, acclimatized in 24 well plates for 30 min, and then fed either an excess of hatched artemia or paramecia. Raw videos of hunting behavior were then recorded for 10 min at 30 fps using a high-resolution monochrome camera (Basler acA4924) and custom Python-based acquisition software.

### High-resolution behavioral tracking

We developed a system (*Johnson et al., 2019*) in which a high-speed infrared camera moves on motorized rails to automatically track a zebrafish larvae in a large pool (300 × 300×4 mm). A single fish is recruited to the arena center with motion cues delivered from a projector to initiate each trial. Paramecia are dispersed throughout the middle of the pool. For analysis 60 Hz image frames are centered and aligned. In every frame, the tail was skeletonized and the gaze angle of each eye is calculated. The eyes can each move from around zero degrees (parallel to body-axis) to 40 degrees (converged for hunting). Each bout was then represented as a point in 220-dimensional posture space by accumulating 22 posture measurements (20 tail tangent angles to encode tail shape, and two eye gaze angles) across 10 image frames (~167 ms) from the beginning of each bout. All bouts were then mapped to a 2-D space with t-distributed stochastic neighbor embedding (t-SNE), Four major hunting bout types can be identified from this embedding. Hunts begin with the 'j-turn', and fish follow and advance toward prey objects with 'pursuit' bouts. Hunts end with an 'abort' or a 'strike'. When the fish is not actively involved in a hunt, it explores the arena with 'exploratory' bouts. Fractions of hunting bouts were then compared between fed and food-deprived fish in 3 min time bins over 45 min.

### Statistics

All error bars show mean ± SEM over fish. Significance was reported as follows: *p<0.05. Significance was determined using the non-parametric Wilcoxon signed-rank test for paired data and the Wilcoxon rank-sum test for independent samples. One-tailed tests were performed in cases where there was a prior prediction regarding the direction of change. A one-or two-way ANOVA (Tukey-Kramer correction, MATLAB statistical toolbox) was used in cases where multiple comparisons were involved.

### Code availability

Analysis code used in this manuscript is available at https://github.com/carolinewee/ROIbasedpERKanalysis (*Wee, 2019a*; copy archived at https://github.com/elifesciences-

publications/ROIbasedpERKanalysis), https://github.com/carolinewee/gutfluorescence (*Wee, 2019b*; copy archived at https://github.com/elifesciences-publications/gutfluorescence) and https://github.com/carolinewee/CellularpERKanalysis (*Wee, 2019c*; copy archived at https://github.com/elifesciences-publications/CellularpERKanalysis).

## Acknowledgements

We thank Harold Burgess for kindly providing the y333:Gal4 transgenic line, and Adam Douglass who provided us with the TH2:GCaMP5 transgenic line. We further thank Thomas Panier who assisted Robert Johnson in construction of the rig used for high resolution behavioral imaging. Support from Steve Turney and the CBS imaging facility, and the Harvard Center for Biological Imaging were essential for the successful completion of many experiments. Finally, we would like to thank Jessica Miller, Steve Zimmerman, Karen Hurley and Brittany Hughes at Harvard for providing invaluable fish care.

## Additional information

### Funding

| Funder | Grant reference number | Author |
| --- | --- | --- |
| National Institutes of Health | Brain Initiative grant U19NS104653 | Florian Engert<br>Sam Kunes |
| National Institutes of Health | Brain Initiative grant R24 NS086601 | Florian Engert |
| National Institutes of Health | Brain Initiative grant R43OD024879 | Florian Engert |
| Simons Foundation | 542973 | Florian Engert |
| Simons Foundation | 325207 | Florian Engert |
| Simons Foundation | 325171 | Misha B Ahrens |
| Simons Foundation | 542943SPI | Misha B Ahrens |
| Agency for Science, Technology and Research | National Science Scholarship (PhD) | Caroline Lei Wee |
| Human Frontier Science Program | LT000626/2016 | Armin Bahl |
| AMED | National BioResource Project | Koichi Kawakami |
| AMED | Fundamental Technologies Upgrading Program | Koichi Kawakami |
| JSPS | JP18H04988 | Koichi Kawakami |

The funders had no role in study design, data collection and interpretation, or the decision to submit the work for publication.

### Author contributions

Caroline Lei Wee, Conceptualization, Data curation, Software, Formal analysis, Supervision, Validation, Investigation, Visualization, Methodology, Writing—original draft, Writing—review and editing; Erin Yue Song, Conceptualization, Data curation, Supervision, Validation, Investigation, Methodology, Writing—review and editing; Robert Evan Johnson, Resources, Data curation, Software, Formal analysis, Validation, Investigation, Visualization, Methodology, Writing—review and editing; Deepak Ailani, Formal analysis, Investigation; Owen Randlett, Software, Formal analysis, Methodology, Writing—review and editing; Ji-Yoon Kim, Investigation; Maxim Nikitchenko, Armin Bahl, Resources, Software; Chao-Tsung Yang, Koichi Kawakami, Resources; Misha B Ahrens, Resources, Software, Supervision, Investigation, Methodology; Florian Engert, Sam Kunes, Conceptualization, Supervision, Funding acquisition, Project administration, Writing—review and editing

## Author ORCIDs

Caroline Lei Wee (iD) https://orcid.org/0000-0003-4599-550X
Erin Yue Song (iD) https://orcid.org/0000-0001-7911-6193
Armin Bahl (iD) http://orcid.org/0000-0001-7591-5860
Misha B Ahrens (iD) http://orcid.org/0000-0002-3457-4462
Koichi Kawakami (iD) https://orcid.org/0000-0001-9993-1435
Sam Kunes (iD) https://orcid.org/0000-0002-2973-8063

## Ethics

Animal experimentation: All protocols and procedures involving zebrafish were approved by the Harvard University/Faculty of Arts & Sciences Standing Committee on the Use of Animals in Research and Teaching (IACUC). Protocol #12-02-2.

## Decision letter and Author response

Decision letter https://doi.org/10.7554/eLife.43775.sa1
Author response https://doi.org/10.7554/eLife.43775.sa2

## Additional files

### Supplementary files

• Supplementary file 1. Z-brain anatomical regions that are more activated in voraciously feeding (food-deprived + food) fish as compared to fed fish.

• Supplementary file 2. Z-brain anatomical regions that are more activated in fed fish as compared to voraciously feeding (food-deprived + food) fish.

• Transparent reporting form

### Data availability

Source data files have been provided for all main figures except for Figure 3. Due to its size, source data for Figure 3 has been uploaded to Dryad (https://doi.org/10.5061/dryad.c610m8n).

The following dataset was generated:

| Author(s) | Year | Dataset title | Dataset URL | Database and Identifier |
|---|---|---|---|---|
| Wee C, Song E, Johnson R, Ailani D, Randlett O, Kim J, Nikitchenko M, Bahl A, Yang C, Ahrens M, Kawakami K, Engert F, Kunes S | 2019 | Data from: A bidirectional network for appetite control in larval zebrafish | https://doi.org/10.5061/dryad.c610m8n | Dryad Digital Repository, 10.5061/dryad.c610m8n |

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

## Appendix 1

### Conceptual Circuit Model

The core issue that arose during the review of the manuscript, and the apparent paradox that manifests in the observant reader, is that cH activity correlates both with Hunger and Satiety - depending on the presence or absence of food. This conundrum is also reflected in the reviewers' comments.

At a more basic level, there was also some concern about usage of words such as hunger and satiety in larval zebrafish, which we believe is largely a semantic issue that is best addressed by providing an operational definition of these terms:

Hunger is an internal state defined by three conditions:

1. The state of being in a caloric/energy deficit - which is usually the consequence of food deprivation
2. A state that may promote food-seeking behavior as well as increased food intake
3. A state that is reflected by - and correlates with - a pattern of modulatory neuronal activity

Voracious feeding is a sub-state of hunger in which food is present and being ingested, but caloric/energy deficit is still high.

Satiety is considered the opposite of hunger and is defined accordingly:

1. A state of having sufficient levels of calories/energy or levels that rest above a homeostatic baseline
2. A state that manifests behaviorally in the slower/lower consumption of food relative to the state of hunger, due to indifference to food or its active avoidance
3. A state that is reflected by an internal modulatory neuronal state that may be antagonistic and opposite to that from hunger

In order to clarify these questions related to our manuscript we give in the following a detailed description of our conceptual model. We describe how this model incorporates the observed activity patterns in all three nuclei (cH, mLH and lLH) and we discuss the role that we propose all three may play in releasing hunting and feeding behavior. We emphasize that this is a working model, and though we have presented partial evidence in support of some aspects, additional studies will be required to conclusively prove and/or refine our hypotheses.

To recap, we observe that cH activity is at baseline when the animal is satiated (well-fed). Activity then increases during food deprivation, and drops to very low levels once food is presented. During feeding, and with the resulting return to satiety, the activity rises back to baseline levels (*Appendix 1—figure 1*).

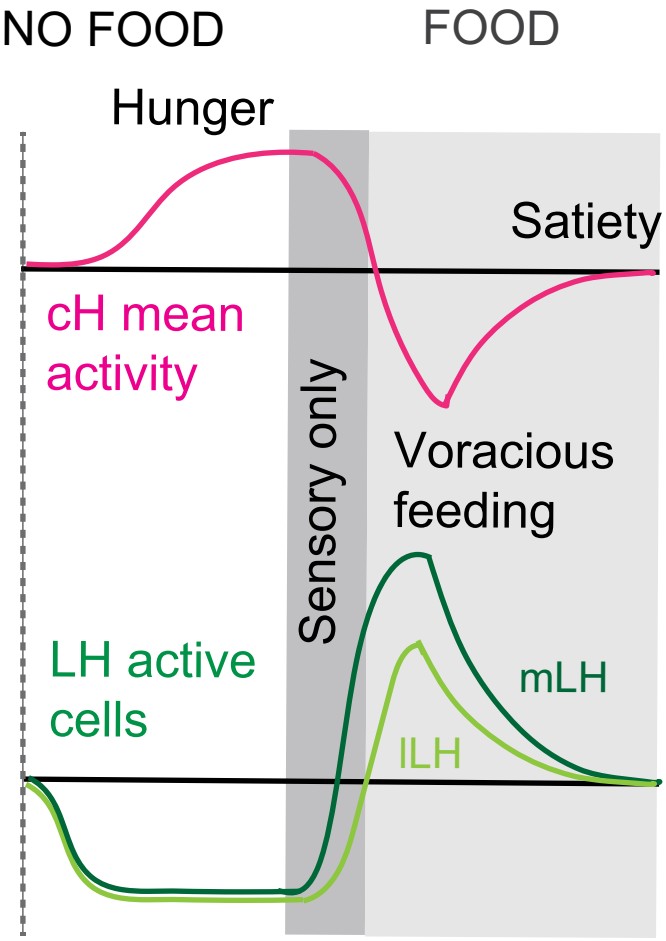

**Appendix 1—figure 1.** Summary of cH and LH activity over Hunger and Satiety. This diagram is the same as in *Figure 4d*. Differential neural activation of the cH and LH regions in response to prey sensation and hunting as compared to prey ingestion.

The general framework of our conceptual circuit model is based on two core assumptions. These are:

1. Similar to mammals and adult fish, **the LH drives food consumption** (*Demski and Knigge, 1971*; *Jennings et al., 2015*; *Roberts and Savage, 1978*). Thus, it makes sense that LH activity *in the presence of food/ingestive cues* is at medium levels during satiety and higher with increased food deprivation time.
2. That the cH and the lLH/mLH have **mutually inhibitory connectivity**, a hypothesis which is backed up by the strong and consistent anti-correlated activity patterns observed in these nuclei. **Further, we now provide evidence demonstrating an inhibitory effect of the cH on lLH activity**.

We will also be making other assumptions which are explicitly outlined below.

Next, we address the seemingly-complex role of the cH, which is complicated by the fact that it has opposite activity patterns depending on whether food is absent or whether it has been detected/ingested. In our manuscript we put forth a number of non-mutually-exclusive hypotheses for the roles of the cH in each of these stages:

1. The cH encodes an aversive, negative valence state, which should induce, when activated in the absence of food, a negative association with the current condition and a drive to change this condition and explore alternatives. Activating cH in the presence of food should reduce

food intake, as observed with other stressors (*De Marco et al., 2014*), and optimize the animal's behavior to remove itself from the negative context. For example in mammals, AgRP neurons, which are activated during hunger, have been shown to encode a similar aversive state (*Betley et al., 2015*). Indeed, there is some evidence in the zebrafish literature that the cH may encode aversive stimuli (*Wee et al., 2019*).

2. The cH drives exploration (food search) as *opposed to* exploitation (food consumption), which could be reflected by enhanced locomotion, increase in search area, enhanced visual acuity or sensitivity to prey-like objects or other subtle changes that might not manifest significantly in spontaneous swimming behavior in zebrafish larva. When activated in the absence of food it might drive enhanced exploration and higher sensitivity to food-like cues. For example in mammals, AgRP neurons promote food seeking and exploratory behaviors (*Dietrich et al., 2015*; *Krashes et al., 2011*). In contrast, activating the cH in the presence of food, while possibly lowering the threshold of initiating food-seeking behavior (i.e. exploration), might ultimately lead to a reduction of food intake and consummation (i.e. exploitation).

3. The cH induces sensitization/priming of the LH circuit such that it is more responsive to future food and ingestive cues. If accurate, when activated in the absence of food it should drive enhanced feeding after food is presented. This is what we have observed and reported in our manuscript (*Figure 6*). If the cH is activated in the presence of food, as long as the priming effect is weaker than the acute cH-mediated inhibition of the LH, driving the cH should simply reduce LH activity and thus also reduce consummatory behavior. Otherwise, if the priming effect is strong, consummatory behavior might be increased. Our experiments activating the cH in the presence of food in both fed and food-deprived fish suggest that any effect of priming is weaker than the acute inhibitory effect of the cH on LH activity.

In the current manuscript, we present partial evidence in support of all three hypotheses. We want to make clear that a conclusive verification of the complete conceptual circuit model (see below) is beyond the scope of this manuscript and will require future studies.

## A putative circuit diagram (*Appendix 1—figure 2*)

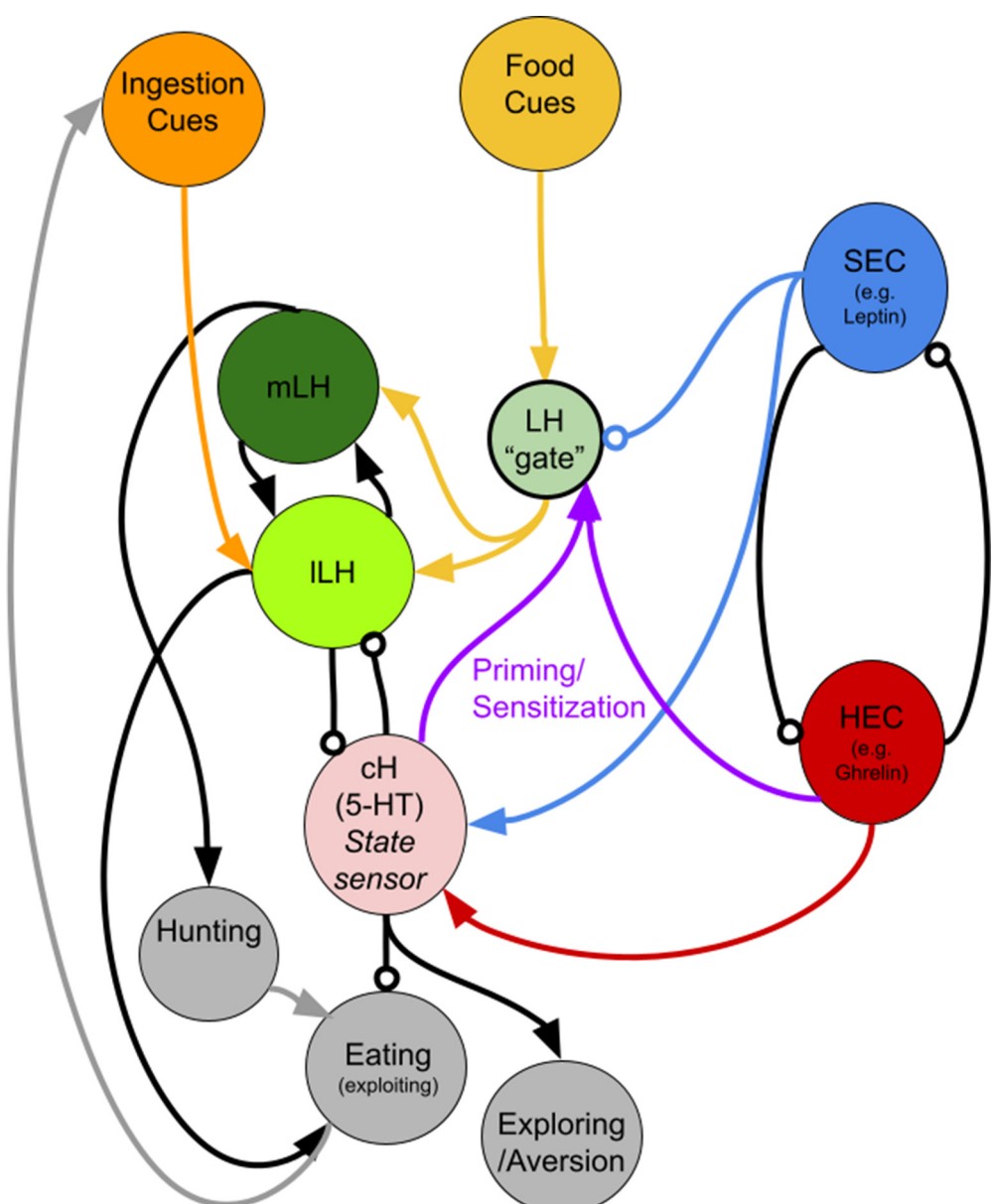

**Appendix 1—figure 2.** A putative circuit diagram HEC = Hunger Encoding Circuit, SEC = Satiety Encoding Circuit, which should have anti-correlated activities and report the animal's energy/caloric status. The cH represents both hunger and satiety state and primes the LH during hunger. It may drive other behaviors such as exploration or aversive behavior, but also suppresses feeding. Other HEC components may also be involved in LH priming/sensitization. We propose mutual inhibition between the cH and LH, though we have only demonstrated unidirectional inhibition (cH on lLH) thus far. The mLH, normally responsive to food cues, may promote hunting, though not necessarily coupled with ingestion, whereas the lLH, which is more responsive to ingestive cues, should enhance further ingestion (i.e. eating). The LH 'gate' is a conceptual representation of how its sensitivity to food cues could be modulated by other signals (i.e. reduced by the SEC and enhanced by cH-mediated priming). It does not necessarily represent a physical neuronal population.

We postulate that the cH receives excitatory input from an undefined source (possibly even directly sensing nutrients/hormones from ventricular cerebrospinal fluid) that generally codes for caloric deficit. We will call this the hunger encoding circuit (HEC). In addition we postulate that the

cH receives excitatory input from a source analogous to the HEC that generally codes for satiety. We will call this unknown circuit the 'satiety encoding circuit' or 'SEC'. It is possible that different neurons within the cH encode each of these cues, but both need to be represented for cH activity to converge stably at 'medium' levels during satiety (where HEC = 0) without drift.

As described before, we propose that the cH also receives inhibitory input from the LH, which is gated by food and ingestive cues, as well as by satiety cues. Thus, in the absence of food, the LH is inactive and the HEC activates the cH, whereas in the presence of food, the cH is initially strongly inhibited by the LH, and then, as the HEC is shut down and the SEC activated, returns to a baseline firing rate.

Finally, on top of this basic circuitry we propose a latent modulatory connection between the cH and LH which primes the sensitivity of the LH during hunger such that it becomes more active once food and ingestive cues are presented. Other yet-to-be-discovered HEC components may also be involved in such priming. Details of the model are outlined below.

Below we discuss in detail the prediction that this model generates for targeted silencing and activation of cH in *food-deprived* as well as *fed* fish in both, the *presence* as well as the *absence* of food (also see *Appendix 1—figure 3*). Note that there are eight possible optogenetic experiments to be conducted with the cH as a target, and they are comprised of three pairs of mutually exclusive conditions: activation vs silencing, food-deprived fish vs fed fish, presence of food vs absence of food. Food intake is currently the clearest behavioral readout, and thus also the focus of this manuscript, though we will continue to pursue experiments in support of other predictions (e.g. exploratory behavior, aversive conditioning).

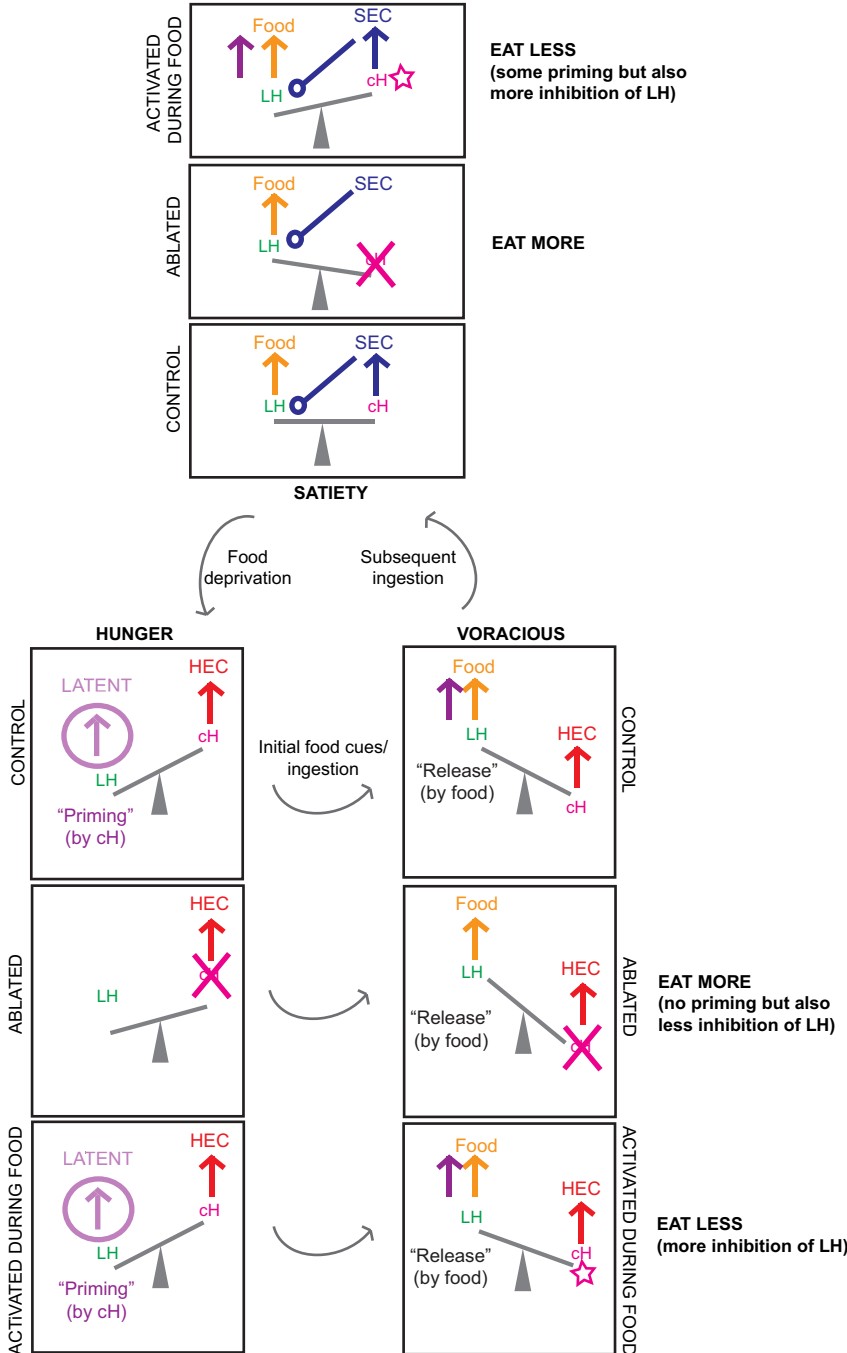

**Appendix 1—figure 3.** Another schematic summarizing our circuit model, including predictions for cH ablation and activation results. Here, the proposed mutual inhibition between cH and LH activity is represented by a 'see-saw', and relevant inputs/outputs during each phase are represented by 'forces'. We represent our predictions for what happens when the cH is ablated, as well as when it is activated in the presence of food, for both satiated and hungry fish. We assume that the effects of priming are weaker than the effects of acute mutual inhibition. Color codes are consistent with *Figure 2*. Note that activation of the cH *prior* to feeding is not depicted in this diagram: in this scenario we predict that this would cause priming of the LH that increases subsequent feeding, the effect of which may be more obvious in satiated fish where cH activity starts off lower (consistent with our optogenetics results).

An additional 16 experiments are possible if the mLH and lLH are also considered possible targets for specific optogenetic perturbation. However, such experiments are currently difficult to do since specific transgenic lines for these regions are not available.

## The serotonergic neurons of the caudal hypothalamus (cH)

We propose that the cH is driven by (i.e. receives excitatory input from) two regulatory centers: the hunger encoding circuit (HEC), and the satiety encoding circuit (SEC), whose existence we postulate, but which we currently cannot back up with supporting experimental data. Thus, the cH can be strongly activated by hunger cues, but also returned to baseline levels by satiety cues after having been strongly suppressed by the LH.

As described above, we postulate that the cH may play multiple roles in regulating feeding behavior. One of these roles may be to 'prime' the LH circuit such that it becomes more active once food is presented. This postulate is supported by the observation that the amount of food cue-induced activity in LH increases with food deprivation time and hence with increased integrated cH activity during food deprivation.

Once food is presented, we posit that the LH, driven by food and ingestive cues, shuts down the cH via inhibitory circuitry. It does so more strongly in food-deprived vs fed fish, as it has been primed by prior cH/HEC activation, and also the SEC is not activated (i.e. the putative 'gate' is wide open). This is supported by evidence that LH activity is much higher after a longer-period of food deprivation (e.g. Manuscript *Figure 2*).

In the presence of food, recovering cH activity levels during feeding lead to a progressive reduction in LH levels and thus reduce food intake.

## Predictions

*Activation* of the cH in the *presence of food* should reduce ingestion and feeding rates. This is consistent with our current optogenetic results in food-deprived fish. In the case of satiated fish (an experiment we did not previously attempt), we predicted that food intake should still be reduced, but counteracted by the effect of priming. We now present results consistent with this hypothesis–food intake trends lower in continuously fed fish, though not significantly.

Activation of the cH in the *absence of food* may lead to 1) avoidance and/or aversive conditioning towards the location at which the cH is being stimulated 2) increased exploratory behavior; for example increase of swim frequency, subtle convergence of eyes, or other subtle changes that might not manifest significantly in spontaneous swimming behavior in zebrafish larva.

Activation of the cH in the *absence of food* may also sensitize the LH circuit to future food cues and enhance food intake. This is consistent with our optogenetic results.

Ablation of the cH should lead to a disinhibition (i.e. a release) of LH activity and thus hunting and ingestion. Though any 'priming' effect would now be reduced, we posit that the acute disinhibition of LH activity would drive up food intake regardless of whether it has been previously 'primed' by the cH. That is, animals should still eat more in the presence of food. This is consistent with our ablation results.

In the absence of food we expect that fish with an ablated cH would display a reduction in exploratory behavior which might manifest in a decrease in swim frequency, divergence of eyes etc; again, this might be hard to pick up because these expected changes are subtle and might not present yet at larval stages.

## The neurons of the lateral part of the lateral hypothalamus (lLH)

We observe that the lLH shows clear anti-correlated activity with the cH. Its activity is at baseline during satiation, activity decreases during food deprivation, switches to high levels when the animal starts feeding, and slowly drops back to baseline level with the return of satiety.

We postulate and now demonstrate using optogenetics that the lLH likely receives inhibitory input from the cH, and that it is also strongly excited by consummatory internal cues, for example food being ingested. An example of such an input is the anterior branch of the esophageal nerve (En2) in Aplysia which is both necessary and sufficient for effective reinforcement of biting behavior (*Brembs et al., 2002*) and which is known to convey information about the presence of food during

ingestive behavior. Based on work from *Muto et al. (2017)*, LH activity (they did not distinguish between the lobes) increases slightly after food detection, but even more strongly immediately after ingestion (*Muto et al., 2017*). We further propose that activation of lLH only occurs if both inputs are activated together (cH inhibition and food ingestive cues) and that its activity drives ingestive behavior (capture swims, biting and swallowing).

## Predictions

Since no specific transgenic lines exist, clean perturbation experiments are not possible; but predictions, of course can be made.

*Activation* of the lLH should lead to voracious feeding in the presence of food.

*Silencing* of the nucleus should shut down consummatory behavior.

## The neurons of the medial part of the lateral hypothalamus (mLH)

We observe that the mLH basically shows the same activity patterns as the lLH. The only difference is that the switch from depression to activation occurs already with the presentation of food cues, though there is also a further enhancement post-ingestion. This is concluded from: 1) our calcium imaging results with live paramecia (which fish are unable to consume); 2) the fact that exposure to artemia as food cues, which fish can hunt but cannot swallow (they are too big) drives activity in mLH but not in lLH.

We also have evidence (not included in the manuscript) that the mLH is responsive to paramecia odor. In addition, evidence of LH activation by vision was also presented by *Muto et al. (2017)*. We thus postulate that the mLH receives strong excitatory input from the sensory modalities that detect food (vision, olfaction etc), and that it is inhibited by the cH.

We further postulate that the mLH drives the transition from exploratory to hunting behavior, that is it drives the initiation of hunting sequences that start with re-orienting turns (j-turns) and eye-convergence and are followed by pursuits of prey.

## Predictions

Similar constraints about the implementation apply, but predictions can be made.

*Activation* should lead to an increased probability of the initiation of hunts vs exploratory swims.

*Silencing* should do the reverse, namely induce a relative reduction in the release of such hunting sequences.

