## [Decision Letter]

Thank you for sending your article entitled "A bidirectional network for appetite control in zebrafish" for peer review at *eLife*. Your article has been evaluated by three peer reviewers, one of whom is a member of our Board of Reviewing Editors, and the evaluation is being overseen by Ronald Calabrese as the Senior Editor.

Summary:

Wee and colleagues describe results from a series of studies where they discovered two anticorrelated populations of hypothalamic neurons regulating feeding behaviors in zebrafish. The manuscript focuses on two brain areas, i.e. caudal (CH) and lateral (LH) hypothalamus, that were differentially activated/suppressed under these conditions. They claim to have associated inverse activity patterns of these areas with states of hunger/satiety. Genetic ablation or optogenetic activation of the caudal hypothalamus (CH) suppressed or enhanced food intake. Because a subset of the CH hypothalamic neurons are serotonergic they imply that serotonergic activity anticipates voracious feeding. The authors argue they have identified a neuronal network, which coordinates energy balance by controlling feeding behavior.

They propose a model in which the LH regulates feeding and the CH acts as a homeostatic regulator of LH to control feeding in the context of a hungry vs. satiated state. The experiments are well designed and use cutting-edge methods to generate observations that support the authors' conclusions. The results are interesting and provide an important advance to what is known about circuits that regulate feeding in fish, and that may be relevant to mammals, but require some additional experiments to support the model. There is also a lack of clarity and detail in many parts of the manuscript that were major distractions.

Essential revisions:

Although this paper presents an interesting story, the rationale and approaches for the experiments are at times difficult to follow and require more detail. Furthermore, they postulate a mutually inhibitory CH, LH circuit (subsection “Mutually opposing hypothalamic networks control zebrafish appetite”), but only show correlative data and don't experimentally test this hypothesis.

The authors use broad terms such as "hunger", "food deprivation", "satiety", "food anticipation" which may be inaccurate or even wrong. In their basic experimental design, they constantly feed the larvae from day 5 to day 7, and then compare the response of larvae that were given excess of paramecia to larvae that were food-deprived for only 2 hours and then re-fed. Whether these conditions represent different states of hunger and/or cognitive response to food (anticipation) as the authors claim is not clear, in particular given the presence of yolk nutrients at these stages. This needs clarification and more precise definition of terms being used.

The claim that CH serotonergic population is associated with anticipated voracious feeding is not convincing. In the case of pERK, the authors present one example that a subset of CH neurons co-localize with 5-HT, however they also show that a different subset is dopaminergic. Without using this marker (5-HT) in all ICA samples one cannot be certain that the active/suppressed neurons are indeed serotonergic. The use of GCaMP expressed in the CH (116A:Gal4) to measure activity (Figure 2) is better, however the authors never show colocalization between 116A:Gal4;UAS:GCaMP and 5-HT. This is important, given the expression variability between the different UAS lines shown by the authors. Hence the authors' conclusion that "serotonergic CH activity reports the extent of food deprivation and anticipates the voracity of future hunting and ingestive behaviours" is not sufficiently supported.

In Subsection “Cellular dissection of hypothalamus neural activity reveals a serotonergic population that anticipates voracious feeding” the authors compare and contrast the activities of CH and LH under different feeding paradigms. However, the data in Figure 1f show CH activity presented as "normalized pERK intensity" but mLH and ILH activity presented as "cell count". In the text the authors state that "the number of active CH cells was dramatically reduced" and cite 1f, which does not show cell count data for CH. Is this a mistake? Unless there is a good reason, CH, mLH and ILH activity levels should all be quantified in the same way, with "normalized pERK intensity" being presumably better than the binary on/off quantification of "cell count". Same issue with Figure 3B.

In Figure 2 the authors interpret the differential GCaMP activity of mLH, lLh and CH as hunger-relevant. However, the data indicates that only mLH is exclusively responsive to food while the other two regions respond to other sensory stimuli, such as water flow. The authors allude to this (subsection “Neuronal activities in the caudal and lateral hypothalamus are anti-correlated over short timescales in response to food sensory cues”) but nevertheless insist on interpreting the data as though it corresponds to feeding/ hunger specifically, rather than "general" sensory processing.

Figure 3: I don't understand the purpose of comparing between artemia and paramecia exposure. The authors state that "artemia are a natural prey that larval fish hunt, but which are too large to consume." Firstly, artemia are saltwater animal and therefore not a natural food source for freshwater zebrafish larvae. Second, while artemia are used as a food source for juvenile (three weeks) and adult fish the authors did not show that introduction of artemia evokes feeding/hunting behavior in larval zebrafish as opposed to other non-food related responses. Therefore, the claim that the mLH response to artemia represents the initial sensory response to food cue is not convincing. Related to this figure the authors again use different parameters to measure neuronal activity, i.e. normalized pERK activity for CH and activated neurons cell count for LH.

The authors use genetic ablation using the UAS:nsfb-mCherry under one CH promoter (116A:Gal4) and optogentic activation using UAS:ReaChR driven by a different promoter (y333:Gal4) and therefore they cannot know if the ablated neurons are the same CH neurons as the light-activated cells. This distinction is crucial, as it is the entire rationale behind performing this dual (optogenetic activation versus chemogenetic ablation) experiment.

The assertion that they specifically ablate or activate serotonergic neurons needs further support as only a small subset of 5-HT neurons is targeted by either nsfb or ReaChR as shown in Figure 1—figure supplement 4.

The y333:Gal4 driver is not well characterized. As the optogentic stimulation is not spatially restricted to the CH the authors should show that ReaChR is not expressed in other brain areas.

As the authors show in Figure 1—figure supplement 4, the 116A:Gal4 CH driver is expressed in other hypothalamic regions (e.g. the PVO) and therefore ablation of other cells may affect food intake.

The authors claim that CH activity causes inhibition of LH (subsection “Functional dissection of cH serotonergic neurons in appetite”) in order to interpret their unexpected result that CH activation in the presence of food (as opposed to prior to the introduction of food) results in reduced food intake. To prove this the authors should show that the optogenetic stimulation of CH neurons will result in decreased LH activity using the 76A:GCaMP measurements.

In Figure 2, what is the evidence that the 76A:Gal4 line is expressed in the same ILH and mLH neurons that are labeled by pERK in Figure 1?

Figure 1G. The axes labels are "ILH" for the Y-axis and "mLH" for the X-axis. What does that mean? What are the units on these axes? Is the "Intensity" quantifying gut fluorescence, # of feeding bouts, or something else? What time point after addition of food is shown in these graphs? Is there statistical analysis to support the idea that LH cell count of food intake is increased as a function of food deprivation time? Why wasn't a similar analysis performed for the CH population? In the text (subsection “Cellular dissection of hypothalamus neural activity reveals a serotonergic population that anticipates voracious feeding”) the authors cite 1g as evidence that both "CH and LH activity are modulated by the length of food deprivation" although no data for CH are shown in 1g.

In Figure 1H, why wasn't the same live imaging analysis performed for LH? The authors later (Figure 2) perform calcium imaging for both CH and LH at the same time, so this should be straightforward. In describing 1h, the authors state "To observe the reciprocal relationships between CH and LH populations in real time, we used calcium imaging to measure the response of CH serotonergic neurons to food deprivation." The ending to the opening "To observe the reciprocal relationships between CH and LH populations in real time,…" should be "…we used calcium imaging to simultaneously measure the response of CH serotonergic and LH neurons to food deprivation.".

The authors state in subsection “Whole brain activity mapping of appetite-regulating regions” that the mammalian analog of the LH has been implicated in appetite control, but is this a good comparison if the zebrafish "LH" does not express the neuropeptides that are thought to regulate feeding in the mammalian LH (i.e. hypocretin, mch, etc)? i.e. should the zebrafish "LH" be given a different name since it appears to contain different neurons, and thus likely has a different function, than the mammalian "LH"? Do the glutamatergic and GABAergic neurons found in the mouse LH co-express the hypocretin, mch, etc. neuropeptides?

Based on the data presented in Figure 3, CH activity appears to be regulated by food consumption as opposed to the sensory cues generated by presence of food, similar to ILH, while mLH activity appears to be inducible by sensory cues alone (although to a lesser extent than sensory cues and food consumption at the same time). In subsection “The activities of cH and LH neurons are differentially modulated by food sensory cues and ingestion” the authors claim that this in accordance with "the strong anti-correlation of CH with lLH activity (compared to mLH activity, Figure 2F)". However, Figure 2D and 2D suggest the opposite, i.e. that cH strongly anti-correlates with mLH and has a mixed correlation/anti-correlation interaction with ILH. Please explain.

Figure 4A suggests that activation of cH in fed animals (which presumably have medium levels of CH acivity, see Figure 1I) prior to the presentation of food, increases subsequent food consumption by mimicking the high levels of cH activity during hunger. The same manipulation in hungry fish has no effect since presumably their levels of CH activity are already high (again see Figure 1I). Figure 4B is more complicated. The authors drive CH neurons during the feeding of food-deprived animals and see a reduction in feeding. This is counter-intuitive; we would expect that high levels of CH activity would induce hunger and hence increase feeding. To explain this, the authors suggest that this manipulation increases CH activity from the low levels seen during feeding (again see Figure 1I) to the medium levels seen during satiety, but not to the highest levels seen during hunger. This is a reasonable hypothesis (although it could be spelled out more clearly, as this is a key point in the manuscript and is not clearly explained). To test this hypothesis the authors should activate CH neurons during the feeding of fed fish. These animals presumably have medium levels of cH activity (see Figure 1I) and thus optogenetic activation should drive CH activity to high levels (as implied for the experiment in Figure 4A). In this case, they should see increased feeding. It is surprising that the authors have not already performed this experiment since that would make Figure 4B symmetric to 4A

Figure 4A,B: Stimulation of CH neurons during feeding results in reduced gut fluorescence. Could this be due to reasons other than specific suppression of feeding? For example, reduced locomotor activity, reduced ability to see prey, and/or impaired ability to execute specific steps in the prey capture sequence? Analysis of prey capture sequence, as previously described by several zebrafish labs, would strengthen the interpretation of this result. Based on text in the methods, it sounds like the data needed for this analysis may have already been collected.

Figure 4. This experiment is missing a control for the possibility that the ReaChR transgenic animals have altered feeding even in the absence of orange light. It would also be useful to show that stimulation of CH neurons is actually achieved using (for example) GCaMP, pERK or cfos. Please also state the genotype of ReaChR- control animals (i.e. do controls contain only the Gal4 or the UAS transgene, or neither?).

Figure 4C: Data showing efficacy of ablation should be shown (i.e. extent and specificity of cell loss in MTZ treated animals), particularly because 2.5 mM MTZ is insufficient to induce robust ablation for most Gal4 lines. Similar to Figure 4A,B, this experiment is lacking a control for the possibility that the transgene affects behavior in the absence of MTZ.

The brain activity response to the presence of paramecia could be a visual-mediated hunting response that could be mimicked by animated paramecia or other sensory responses, such as olfactory-mediated motor response. Both of which could increase the chances of successful prey capture resulting in increased food intake.

[Editors’ note: the revised article was rejected after discussions between the reviewers, but the authors were invited to resubmit after an appeal against the decision.]

Thank you for submitting your work entitled "A bidirectional network for appetite control in zebrafish" for consideration by *eLife*. Your article has been reviewed by a Senior Editor, a Reviewing Editor, and two reviewers. The reviewers have opted to remain anonymous.

Our decision has been reached after consultation between the reviewers. Based on these discussions and the individual reviews below, we regret to inform you that your work will not be considered further for publication in *eLife*.

As you can see, one reviewer raised several remaining substantive concerns. After extensive discussion with both reviewers, we decided that significantly more experiments would be required to address these concerns. Thus, we have decided to reject your submission.

*Reviewer #2:*

I really struggled to read and understand this revised manuscript. This is particularly disappointing because many of the criticisms of the first version of the manuscript were related to a lack of clarity and details, and if anything, the revision is worse. I commend the authors for adding significant new experiments. However, most of these experiments are poorly described, appear to contain mistakes, and often cannot be evaluated. There is a general lack of rigor and quantification of key measures, including measures that were requested by reviewers, with only general statements about observations, and in some cases improper comparisons.

Essential revisions:

1) In the text, when the authors cite a supplemental figure, most times they do not say which panel they are referring to. This might seem like a trivial issue, but it eventually makes reviewing/reading the paper difficult. This problem is particularly acute for Figure 2—figure supplement 4 and Figure 2—figure supplement 5. These figures are quickly mentioned and not explained at all in the main text or figure legends. Other supplemental panels (e.g. Figure 2—figure supplement 1A and 1C) show data that seem not to be covered in the text. Please, in the main text, reference every panel of every figure, carefully explain the experiments and describe what they show.

2) Figure 1—figure supplement 3: It is inappropriate to generate ICA data using fish from other feeding-related treatments that are not described in this manuscript. While their inclusion may be necessary to achieve statistically significant results, this is a big black box of data that cannot be evaluated by reviewers or readers. This undescribed data should either be added to the manuscript, or the ICA analysis must be removed.

3) Figure 2: The data presented in this figure suggest that 30 minutes of food deprivation is enough to cause a shift in ILH activity (when quantified by active cell count) but not in cH (when that is quantified by normalized pERK intensity). However, when ILH activity is quantified by normalized pERK intensity, 30 minutes of food deprivation is not enough to cause a shift in activity. In their response to the reviews, the authors noted that normalized pERK intensity is less precise than active cell count; perhaps if cH activity was quantifiable through cell count, 30 minutes would be enough to cause a change in that population as well. Regardless, the authors should not compare the timelines of cH and LH activity changes when using different metrics to quantify the activity of each group (as they do in subsection “Satiation state influences the sensitivity of cH and LH populations to food”, subsection “Mutually opposing hypothalamic networks control zebrafish appetite” and Figure 2D).

4) Subsection “Satiation state influences the sensitivity of cH and LH populations to food”: The text here is not justified by the data presented. The idea that the cH response to absence of food, which the authors claim happens after the LH response, is somehow required for LH responsiveness, does not make sense.

5) Figure 2—figure supplement 1A: What is the feeding condition for these graphs? What message are they intended to convey, particularly because the correlations are relatively weak (especially for cH).

6) Figure 2—figure supplement 2A is problematic. The top and right images are at different scales, and the two pERK images looking very different. There seem to be much fewer TH2 positive cells than in other images the authors provide (e.g. panel e of same figure). A full z-stack of the cH area should be provided.

7) Figure 2—figure supplement 3D: "Consistent with our pERK results, the initial calcium-mediated mean fluorescence and firing frequency of a subset of cH neurons scaled with the length of food deprivation prior to imaging (Figure 2—figure supplement 3D)". There are two problems with this statement. First, this data only shows absolute fluorescence (or is it mean fluorescence?), not firing frequency. Second, what is this "subset of cH neurons"? How many cells are quantified? Where are they? Is this a small minority of the cells, or a general feature of most cH neurons? The relevant cells must be indicated.

8) Figure 2—figure supplement 4 and Figure2—figure supplement 5: There is almost no description or explanation of the data shown in these figures, making them completely incomprehensible to this reviewer. As best as I can understand this data, there appear to be several conflicts between what is shown in different panels, with the pERK data, and what is briefly stated in the text. Maybe it's obvious for aficionados, but likely not for most readers. There are also several apparent problems. First, Figure 2—figure supplement 4 panels A and B show a blue line in the bottom line graphs – what does this correspond to? What do the different numbers indicate – # of neurons? The authors imply opposing activity patterns for lLH and cH based on Figure 2—figure supplement 4A and B (although they are not anti-correlated, just shifted relative to each other), and this relationship is not apparent in the data shown in Figure 2—figure supplement 5. These figures simply cannot currently be evaluated.

9) Related to the last point, subsection “Satiation state influences the sensitivity of cH and LH populations to food”: "While some mLH and lLH voxels showed a predicted reduction in baseline fluorescence and firing rate, many others displayed a significant enhancement of baseline activity." It seems (Figure 2—figure supplement 4B) that most of the LH neurons (especially in the ILH) show increased activity during food deprivation using live imaging, which is the opposite of the pERK results shown in Figures1 and Figure 2 and the outline shown in Figure 2D. Unless I am misunderstanding Figure 2—figure supplement 4 (which is quite possible), this is a major problem that is glossed over. Should the authors focus on the GCaMP results as opposed to the pERK "Given the indirect nature of activity mapping in post-fixed animals"?

10) Figure 4: Text describing this figure states: "exposure to this food cue in the absence of ingestion induced a small increase in lLH neural activity and a larger increase in mLH activity (Figure 4A,B). The artemia-induced hypothalamic activity was, however, less than that observed with consumable prey (Figure 4A/B)." This statement accurately describes the data. However, the next sentence: "These observations suggest that the mLH responds primarily to sensory cues and/or induced hunting behavior whereas the induction of lLH activity largely depends on consumption" is not an appropriate interpretation of the data. I would conclude that both lLH and mLH respond to both paramecia and artemia, that both populations are less responsive to artemia, and that ILH is less responsive than mLH to both stimuli, rather than that one population responds primarily to sensory cues while the other population responds to consumption. One could argue that the increase in cell counts in lLH in response to artemia is very small, however a statistically significant difference is indicated. The "active cell count" metric also seems to be flawed because the mLH area is much larger than the lLH area, and thus these values must be normalized in order to make any meaningful comparisons between the cell populations. It looks to me like normalization would likely eliminate any difference in active cell count between lLH and mLH. It is unclear if the "normalized pERK intensity" metric is similarly flawed, i.e. does this quantify the total fluorescence in the region of interest (which would be affected by the size of the region), or is this value normalized according to the area of measurement (it's unclear what "normalized" refers to here – normalized to tERK, to total area, to the control value)?

11) Figure 5E: How do the authors account for the increase in mLH GCaMP fluorescence in response to stimulation of a control area that is not labeled by ReaChR?

12) Figure 5—figure supplement 1: As the authors note, the y333:Gal4 line is much less specific for 5-HT neurons than the 116A:Gal4 line. It is essential to determine whether the ~40% of y333:Gal4 expressing cells that are 5-HT negative are TH2 positive, as was done for the 116A:Gal4 line, since this would provide a significant dopaminergic input to this experiment.

13) Figure 6—figure supplement 1: This is another example of anecdotal evidence that should be quantified.

14) Figure 6—figure supplement 2: There is no quantification of cH cell ablation. Instead, just a single exemplar image is shown. It is therefore impossible to draw any conclusions about whether or not, or to what degree, the ablation was successful. Thus, the authors' claim in the rebuttal that "We can absolutely confirm that this protocol (which we have also utilized for other transgenic lines) is sufficient to ablate most cH neurons (see Figure 6—figure supplement 2)" is in no way substantiated by the data provided.

15) The authors propose that "optogenetic stimulation of cH activity inhibits lLH activity and thereby causes the feeding rate to decrease." (subsection “Functional dissection of cH serotonergic neurons in feeding behaviour”). They provide evidence that cH activation inhibits feeding (although that needs to be clarified; see comment below), and that cH activation also inhibits ILH activity. Is there any causal evidence that reduced ILH activity reduces feeding? If not, this statement should be deleted.

16) Despite the caveats mentioned by the authors, the use of different Gal4 lines with different expression patterns (which remain lightly characterized despite reviewer requests for more details) for genetic ablation and optogenetic activation precludes the authors from drawing firm conclusions from these experiments. It doesn't really make sense that a Gal4 line would be strong enough to drive cell ablation (especially since the authors use an unusually low concentration of mtz) but not optogenetic stimulation. Just because a Gal4 line doesn't produce the hoped for phenotype (for which the data is not shown) does not mean that one can simply substitute another Gal4 line that does produce the hoped for result.

*Reviewer #3:*

The authors have addressed the majority of my comments to the extent that I now support publication in e*Life*.

---

## [Author Response]

Essential revisions:Although this paper presents an interesting story, the rationale and approaches for the experiments are at times difficult to follow and require more detail.

We have revised the text with an emphasis on clarity and more detail when necessary. We have also enclosed a supplementary document (Conceptual_Circuit_Model.pdf) that explains our overall model.

Furthermore, they postulate a mutually inhibitory CH, LH circuit (subsection “Mutually opposing hypothalamic networks control zebrafish appetite”), but only show correlative data and don't experimentally test this hypothesis.

It is true that we had previously not performed targeted perturbation to test specifically for the mutually inhibitory circuit. We thus attempted to address this question using targeted optogenetic activation of the cH and simultaneous calcium imaging, and present the results in new Figure 5. Specifically, we show that optogenetic activation of the cH using *Tg(y333:Gal4;UAS:ReaChR-RFP)* causes sustained inhibition of the lLH, but not the mLH, unlike stimulation of a nearby control region.

The authors use broad terms such as "hunger", "food deprivation", "satiety", "food anticipation" which may be inaccurate or even wrong. In their basic experimental design, they constantly feed the larvae from day 5 to day 7, and then compare the response of larvae that were given excess of paramecia to larvae that were food-deprived for only 2 hours and then re-fed. Whether these conditions represent different states of hunger and/or cognitive response to food (anticipation) as the authors claim is not clear, in particular given the presence of yolk nutrients at these stages. This needs clarification and more precise definition of terms being used.

We have defined in the text a formal definition of hunger and satiety, and have tried t*o* mostly restrict its usage to the introduction and discussion. We also would like to note that we have previously published work (Jordi et al., 2015, 2018) related to appetite regulation in zebrafish that utilizes a similar experimental design. Further, we are assaying behavior at 7-8 dpf, a stage at which yolk nutrients are largely depleted (Gut et al., 2013).

Here we also define the terms:

Hunger is an internal state defined by three conditions:

1) The state of being in a nutrient/energy deficit – which is usually the consequence of food deprivation or starvation.

2) A state that may promote food-seeking behavior as well as increased food intake.

3) A state that is reflected by – and correlates with – a pattern of modulatory neuronal activity.

Voracious feeding is a sub-state of hunger in which food is present and being ingested, but nutrient/energy deficit is still high.

Satiety is considered the opposite of hunger and is defined accordingly:

1) A state of having sufficient levels of nutrient/energy or levels that rest above a homeostatic baseline.

2) A state that manifests behaviorally in the slower/lower consumption of food relative to the state of hunger, due to indifference to food or its active avoidance.

3) A state that is reflected by an internal modulatory neuronal state that may be antagonistic and opposite to that from hunger.

Of note, none of these definitions require that the fish experiences feelings of hunger/or a “desire” to eat as humans experience it, and we agree with reviewers that the resultant changes in food intake/behavioral output could be due to modulation of sensitivity towards food cues and/or the probability of prey capture success, without invoking additional mechanisms.

Food anticipation: We apologize for the confusion. Our use of the word “anticipation” was simply to describe a neuronal state that precedes and predicts future behavior. Thus, since both natural (i.e. during food deprivation) and artificial (i.e. optogenetic) cH activation occurs before subsequent voracious feeding, we described it as “anticipation”.

We do realize that “anticipation” could be easily interpreted to mean a cognitive expectation of future food and/or imply preparatory behavior, neither of which we claim the zebrafish to be doing. Thus, we have removed it and replaced it with clearer terminology (e.g. “priming” or “sensitization”), as we have done in our discussion of putative roles of the cH in appetite control (please also refer to the supplementary document: Conceptual_Circuit_Model.pdf for more details).

The claim that CH serotonergic population is associated with anticipated voracious feeding is not convincing. In the case of pERK, the authors present one example that a subset of CH neurons co-localize with 5-HT, however they also show that a different subset is dopaminergic. Without using this marker (5-HT) in all ICA samples one cannot be certain that the active/suppressed neurons are indeed serotonergic.

We have now made it clear that we are referring to the cH (not the serotonergic cH) in low-resolution pERK experiments (MAP-mapping) and ICA analysis. We also would like to emphasize that these coarse pERK experiments serve primarily to narrow down regions of interest and focus our attention on subsequent, more detailed, circuit dissection with a combination of calcium imaging, optogenetic activation and chemical ablation.

The use of GCaMP expressed in the CH (116A:Gal4) to measure activity (Figure 2) is better, however the authors never show colocalization between 116A:Gal4;UAS:GCaMP and 5-HT. This is important, given the expression variability between the different UAS lines shown by the authors. Hence the authors' conclusion that "serotonergic CH activity reports the extent of food deprivation and anticipates the voracity of future hunting and ingestive behaviours" is not sufficiently supported.

We have now also demonstrated that the majority of neurons labeled by our Gal4 lines are serotonergic. Specifically, we have quantified the overlap of *Tg(116A:Gal4)* neurons with 5-HT staining to be 88.9 ± 0.8%, and the *Tg(y333:Gal4;UAS:ReaChR-RFP)* line to be 57.4 ± 2.1%. As such, we are confident that at least a subset of serotonergic cH neurons are modulated by food deprivation and food cues (i.e. Figure 2—figure supplement 3, Figure 2—figure supplement 4 and Figure 2—figure supplement 5, Figure 3). At the same time, we have now made it explicit in the text that we do not rule out the role of additional neuromodulators, particularly dopamine.

In Subsection “Cellular dissection of hypothalamus neural activity reveals a serotonergic population that anticipates voracious feeding” the authors compare and contrast the activities of CH and LH under different feeding paradigms. However, the data in Figure 1f show CH activity presented as "normalized pERK intensity" but mLH and ILH activity presented as "cell count". In the text the authors state that "the number of active CH cells was dramatically reduced" and cite 1f, which does not show cell count data for CH. Is this a mistake? Unless there is a good reason, CH, mLH and ILH activity levels should all be quantified in the same way, with "normalized pERK intensity" being presumably better than the binary on/off quantification of "cell count". Same issue with Figure 3B.

In the LH (unlike the cH), activated cells are scattered, leaving clear units (i.e. concentrated regions of higher fluorescence) that can be identified by automated thresholding (see new Figure 1—figure supplement 4). Thus, for the LH we used cell count rather than mean fluorescence, as it provides higher-resolution information about activity as compared to averaging the fluorescence over the entire region, and is less susceptible to staining variability. However, we agree with the referees that different metrics can be confusing, and that increased activity also happens in the neuropil. Thus, we have also analyzed our data in terms of absolute fluorescence and present both metrics for readers. Note that fluorescence tends to have larger variability than cell count. We have also changed the text to distinguish between when referring to cell counts vs overall fluorescence.

In Figure 2 the authors interpret the differential GCaMP activity of mLH, lLh and CH as hunger-relevant. However, the data indicates that only mLH is exclusively responsive to food while the other two regions respond to other sensory stimuli, such as water flow. The authors allude to this (subsection “Neuronal activities in the caudal and lateral hypothalamus are anti-correlated over shorttimescales in response to food sensory cues”) but nevertheless insist on interpreting the data as though it corresponds to feeding/ hunger specifically, rather than "general" sensory processing.

We certainly do not mean to claim, neither do we believe, that feeding/appetite control is the only role for the cH and the lLH (or even the mLH). We have now explicitly stated this in the discussion to be clearer about our interpretations.

Figure 3: I don't understand the purpose of comparing between artemia and paramecia exposure. The authors state that "artemia are a natural prey that larval fish hunt, but which are too large to consume." Firstly, artemia are saltwater animal and therefore not a natural food source for freshwater zebrafish larvae. Second, while artemia are used as a food source for juvenile (three weeks) and adult fish the authors did not show that introduction of artemia evokes feeding/hunting behavior in larval zebrafish as opposed to other non-food related responses. Therefore, the claim that the mLH response to artemia represents the initial sensory response to food cue is not convincing. Related to this figure the authors again use different parameters to measure neuronal activity, i.e. normalized pERK activity for CH and activated neurons cell count for LH.

We have removed the word “natural” from the sentence. Since juvenile and adult fish will readily hunt and eat artemia, we believe that the fact that they are not a natural prey is not too big of an obstacle for our interpretation. We now also provide video recordings showing hunting (J-turns and pursuit bouts) of artemia by larval zebrafish, despite being too large to consume.

The appeal of using artemia is that larval fish readily pursue and hunt them, but that they cannot swallow them because they are too big. As such this allows a clear demonstration that all hunting related sensory experience (including the re-afferent experience of the hunts itself) is insufficient to drive high lLH and mLH activity and to suppress cH. Apparently, it is only the act of ingestion/swallowing that induces this switch. We have more clearly described this rationale in the text.

We have also analyzed the pERK data using the same parameters (i.e. mean fluorescence) for both the cH and LH.

The authors use genetic ablation using the UAS:nsfb-mCherry under one CH promoter (116A:Gal4) and optogentic activation using UAS:ReaChR driven by a different promoter (y333:Gal4) and therefore they cannot know if the ablated neurons are the same CH neurons as the light-activated cells. This distinction is crucial, as it is the entire rationale behind performing this dual (optogenetic activation versus chemogenetic ablation) experiment.

We share the reviewer’s concern over use of the two transgenic lines. Multiple unsuccessful attempts were made to robustly express UAS-channelrhodopsin variants with the *Tg(116A:Gal4)* line. Ultimately, we used the *Tg(y333:Gal4)* transgenic line for optogenetics, which we have now quantified to show significant (57.4 ± 2.1%) overlap with 5-HT in the cH, and we also verify that this line has relatively specific expression in the cH and PVO expression (Figure 5—figure supplement 1).

As such we can conclude that the ablated neurons likely share significant overlap with the light activated ones. However, we also explicitly discuss this caveat in the text.

The assertion that they specifically ablate or activate serotonergic neurons needs further support as only a small subset of 5-HT neurons is targeted by either nsfb or ReaChR as shown in Figure 1—figure supplement 4.

We have toned down on our claims that we are specifically ablating/activating serotonergic neurons, however, we note that the *Tg(116A:Gal4)* line is ~90% serotonergic and has minimal overlap with dopaminergic neurons (Figure 2—figure supplement 2).

The y333:Gal4 driver is not well characterized. As the optogentic stimulation is not spatially restricted to the CH the authors should show that ReaChR is not expressed in other brain areas.

We have shown in Figure 5—figure supplement 1 using whole-mount imaging that *Tg(y333:ReaChR-RFP)* expression is quite specific to the cH. However, there appears to also be labeling of some olfactory bulb neurons as well as some scattered neuron labeling in other parts of the brain. We have noted these possible caveats in the text and Discussion section.

As the authors show in Figure 1—figure supplement 4, the 116A:Gal4 CH driver is expressed in other hypothalamic regions (e.g. the PVO) and therefore ablation of other cells may affect food intake.

The reviewers are right that the *Tg(116A:Gal4)* line labels neurons in both the cH and PVO. However, given that *Tg(UAS:nfsb-mCherry)* expression is particularly weak relative to *Tg(UAS:GFP)* expression in the PVO (we estimate 6-8 cells in the aPVO, 2-4 cells in pPVO, and 30-40 cells in the cH), it is unlikely that there would be substantial PVO ablation (see Figure 6—figure supplement 2).

For *Tg(y333:Gal4;UAS:ReaChR-RFP)* optogenetic experiments (Figure 6), stimulation of the PVO during free-swimming behavior is unavoidable, which we have discussed in the text. However, in our optogenetics + calcium imaging experiments (Figure 5) we have specifically targeted the cH, and shown that its activation is sufficient to suppress LH activity.

It is certainly possible though that, if the right circuit connectivity exists, the PVO could also be indirectly activated. Overall, we do not rule out a role of the serotonergic PVO neurons in feeding, as many medially-situated neurons besides the cH are activated during food deprivation. Again, we have raised this possibility in the text.

The authors claim that CH activity causes inhibition of LH (subsection “Functional dissection of cH serotonergic neurons in appetite”) in order to interpret their unexpected result that CH activation in the presence of food (as opposed to prior to the introduction of food) results in reduced food intake. To prove this the authors should show that the optogenetic stimulation of CH neurons will result in decreased LH activity using the 76A:GCaMP measurements.

The reduction of food ingestion after cH activation is not unexpected on the basis of our understanding of cH and LH activity patterns (see Conceptual_Circuit_Model.pdf for more details). We have now confirmed using optogenetics that cH activation indeed reduces lLH activity (Figure 5).

In Figure 2, what is the evidence that the 76A:Gal4 line is expressed in the same ILH and mLH neurons that are labeled by pERK in Figure 1?

We have now used pERK staining to measure activity after providing food stimuli to food-deprived *Tg(76A:Gal4;UAS-GCaMP6s)* transgenic animals. As far as we can tell, the *Tg(76A:Gal4)* line appears to comprehensively label all LH neurons, and all pERK-positive cells induced by food appear to be double-labeled by *Tg(76A:Gal4;UAS:GCaMP6s) (*Figure 2—figure supplement 3).

Figure 1G. The axes labels are "ILH" for the Y-axis and "mLH" for the X-axis. What does that mean? What are the units on these axes? Is the "Intensity" quantifying gut fluorescence, # of feeding bouts, or something else? What time point after addition of food is shown in these graphs? Is there statistical analysis to support the idea that LH cell count of food intake is increased as a function of food deprivation time? Why wasn't a similar analysis performed for the CH population? In the text (subsection “Cellular dissection of hypothalamus neural activity reveals a serotonergic population that anticipates voracious feeding”) the authors cite Figure 1G as evidence that both "CH and LH activity are modulated by the length of food deprivation" although no data for CH are shown in Figure 1G.

We apologize for the lack of clarity. In the main figure (new Figure 2), we now show a more comprehensive plot (comprising an independent dataset) showing statistically significant changes in both cH and LH activity as a function of food deprivation time.

We have furthermore added mean fluorescence measurements of the cH and LH lobes, and statistical quantification to this original dataset (originally Figure 1G), which we have now moved to new Figure 2—figure supplement 1. To clarify: the axes refer to the number of active (i.e. pERK-positive cells) cells; we have noted this accordingly and also measured average fluorescence. Intensity refers to gut fluorescence intensity, that is, an approximation of food intake. The time point shown is 15 minutes after food addition -- all this information is now described in the legends.

In Figure 1H, why wasn't the same live imaging analysis performed for LH? The authors later (Figure 2) perform calcium imaging for both CH and LH at the same time, so this should be straightforward.

The cH experiments were initially performed before we were aware of the existence of the LH line. Currently, as the reviewer suggests, we have now performed simultaneous monitoring of the cH and LH over the course of food deprivation using the 116A and 76A transgenic lines, presented in the new Figure 2—Figure Supplement 4 & Figure 2—figure supplement 5.

Briefly, we confirmed that a subset of cH neurons (and also average cH activity) increases over the course of food deprivation, consistent with our pERK results. However, whereas from our pERK data we find that the number of active cells in the mLH and lLH (but not mean pERK fluorescence), is reduced after food deprivation (Figure 2), our calcium imaging results reveal large subsets of LH voxels that increase in baseline fluorescence over the course of food deprivation.

The changes in LH activity may reflect responses to head-fixation. Alternatively, we hypothesize that over the course of food deprivation, the LH is being sensitized by the cH and/or other hunger-related cues, which could explain the subsequent enhanced response to food and food cues. Notably, unlike “active cell count”, the mean LH fluorescence is not clearly reduced over the course of food deprivation, leaving open the possibility of subthreshold increases in cellular or neuropil activity. We have discussed both possibilities in the text. Finally, lLH calcium spikes still were on average accompanied by a reduction in cH fluorescence, suggesting that these loci still maintain an anti-correlated relationship over food deprivation.

We have now moved Figure 1H to Figure 2—figure supplement 3.

In describing Figure 1H, the authors state "To observe the reciprocal relationships between CH and LH populations in real time, we used calcium imaging to measure the response of CH serotonergic neurons to food deprivation." The ending to the opening "To observe the reciprocal relationships between CH and LH populations in real time,…" should be "…we used calcium imaging to simultaneously measure the response of CH serotonergic and LH neurons to food deprivation.".

We thank the reviewer for their suggestion and have made appropriate modifications.

The authors state in subsection “Whole brain activity mapping of appetite-regulating regions” that the mammalian analog of the LH has been implicated in appetite control, but is this a good comparison if the zebrafish "LH" does not express the neuropeptides that are thought to regulate feeding in the mammalian LH (i.e. hypocretin, mch, etc)? i.e. should the zebrafish "LH" be given a different name since it appears to contain different neurons, and thus likely has a different function, than the mammalian "LH"? Do the glutamatergic and GABAergic neurons found in the mouse LH co-express the hypocretin, mch, etc. neuropeptides?

We and others propose the fish LH to be homologous to the mammalian LH due to broad anatomical and functional similarities determined by stimulation, ablation and more lately, also imaging experiments (Demski, 1973; Muto et al., 2017; Roberts and Savage, 1978). We do agree that its known neuromodulatory phenotype evidently does not overlap with that of mammals, however, since other neuronal types in the mammalian LH (e.g. GABAergic neurons that are neither MCH nor Orexin-positive) have been shown to be involved in both food responses and feeding behavior (Jennings et al., 2015), we believe that there may be some unique and conserved feeding-related functions ascribed to the LH that are independent of MCH and Orexin. Interestingly, a recent study (Mikelsen et al., 2019) has identified additional neuromodulators/peptides co-expressed with LH GABAergic and glutamatergic neurons in mice, and that cluster into further subpopulations, which may form the basis of future investigations. Overall, we are inclined to stick with the same terminology, but have tried to be clear in our text about the differences between the fish and mammalian LH.

Based on the data presented in Figure 3, CH activity appears to be regulated by food consumption as opposed to the sensory cues generated by presence of food, similar to ILH, while mLH activity appears to be inducible by sensory cues alone (although to a lesser extent than sensory cues and food consumption at the same time). In subsection “The activities of cH and LH neurons are differentially modulated by food sensory cues and ingestion” the authors claim that this in accordance with "the strong anti-correlation of CH with lLH activity (compared to mLH activity, Figure 2F)". However, Figure 2D and 2D suggest the opposite, i.e. that cH strongly anti-correlates with mLH and has a mixed correlation/anti-correlation interaction with ILH. Please explain.

We note that the results may appear confusing. It can be explained by the fact that while there are fewer anti-correlated voxels between the cH and the lLH, the magnitude of anti-correlation (i.e. r-value) for these voxels tend to be stronger relative to those in the mLH. In contrast, the mLH has many, but less strongly anti-correlated voxels with the cH. We have revised the text to include this explanation.

Figure 4A suggests that activation of cH in fed animals (which presumably have medium levels of CH acivity, see Figure 1I) prior to the presentation of food, increases subsequent food consumption by mimicking the high levels of cH activity during hunger. The same manipulation in hungry fish has no effect since presumably their levels of CH activity are already high (again see Figure 1I). Figure 4B is more complicated. The authors drive CH neurons during the feeding of food-deprived animals and see a reduction in feeding. This is counter-intuitive; we would expect that high levels of CH activity would induce hunger and hence increase feeding. To explain this, the authors suggest that this manipulation increases CH activity from the low levels seen during feeding (again see Figure 1I) to the medium levels seen during satiety, but not to the highest levels seen during hunger. This is a reasonable hypothesis (although it could be spelled out more clearly, as this is a key point in the manuscript and is not clearly explained). To test this hypothesis the authors should activate CH neurons during the feeding of fed fish. These animals presumably have medium levels of cH activity (see Figure 1I) and thus optogenetic activation should drive CH activity to high levels (as implied for the experiment in Figure 4A). In this case, they should see increased feeding. It is surprising that the authors have not already performed this experiment since that would make Figure 4B symmetric to 4A.

Reduced food intake as a result of cH activation was not an unexpected result to us (see Conceptual_Circuit_Model.pdf for detailed explanation). In the presence of food cH activity rises with increasing satiety, which is anti-correlated with LH activity, so we expected that as such increasing cH activity would be associated with lower LH activity and reduced food intake.

Thus, we predicted that activation of the cH during feeding in fed fish would still likely reduce food intake, though the degree of cH-induced “priming” of LH circuitry may also affect the results. As suggested by the reviewer, we have now performed this experiment. While we have a relatively low sample size due to current difficulties in breeding our transgenic fish, the results trend towards a suppression in feeding, and do not support the idea that cH stimulation in fed fish can increase food intake. This, along with optogenetic imaging experiments confirms our model that the acute effect of cH stimulation is a reduction of food intake likely via suppression of the lLH, whereas stimulation of the cH in the absence of food “primes” or “sensitizes” LH circuitry to enhance subsequent feeding.

Figure 4A,B: Stimulation of CH neurons during feeding results in reduced gut fluorescence. Could this be due to reasons other than specific suppression of feeding? For example, reduced locomotor activity, reduced ability to see prey, and/or impaired ability to execute specific steps in the prey capture sequence? Analysis of prey capture sequence, as previously described by several zebrafish labs, would strengthen the interpretation of this result. Based on text in the methods, it sounds like the data needed for this analysis may have already been collected.

This is a very good idea; however, we unfortunately have not performed experiments in which we monitor behavior while the cH is being activated during feeding, as this is technically difficult to do at high throughput as compared to measuring gut fluorescence. In these particular experiments, the LED is placed directly above the dish, precluding video analysis, though we hope to utilize a more sophisticated setup in future experiments.

Figure 4. This experiment is missing a control for the possibility that the ReaChR transgenic animals have altered feeding even in the absence of orange light. It would also be useful to show that stimulation of CH neurons is actually achieved using (for example) GCaMP, pERK or cfos. Please also state the genotype of ReaChR- control animals (i.e. do controls contain only the Gal4 or the UAS transgene, or neither?).

We have now included in Figure 6—figure supplement 1 some evidence using pERK staining that our optogenetic manipulation for free-swimming behavior is sufficient to activate ReaChR-positive cells, particularly the ones with the strongest expression. We also have shown using calcium imaging that ReaChR activation does lead to expected increases in GCaMP fluorescence, though this is using a different setup (Figure 5).

Controls for the ReaChR experiments do not have visible *Tg(y333:Gal4;UAS:ReaChR-RFP)* expression, and thus are a mixture of siblings expressing *Tg(y333:Gal4) only, Tg(UAS:ReaChR-RFP)* or neither of these transgenes, each with one third probability. We have now stated this explicitly, also for *Tg(116A:Gal4)*, in the text.

Given that depending on the timing of light stimulation (before or during food), *Tg(y333:Gal4;UAS:ReaChR-RFP)* fish either increase or decrease their feeding, it is unlikely that the transgene itself affects behavior in a systematic way. We have been having issues generating sufficient ReaChRpositive embryos to do these and other requested experiments, and therefore did not manage to perform this particular control in time. However, we have performed transgene-only controls for the ablation experiment (see below).

Figure 4C: Data showing efficacy of ablation should be shown (i.e. extent and specificity of cell loss in MTZ treated animals), particularly because 2.5 mM MTZ is insufficient to induce robust ablation for most Gal4 lines. Similar to Figure 4A,B, this experiment is lacking a control for the possibility that the transgene affects behavior in the absence of MTZ.

We can absolutely confirm that this protocol (which we have also utilized for other transgenic lines) is sufficient to ablate most cH neurons (see Figure 6—figure supplement 2). We also have data showing that the transgene alone does not affect behavior in the absence of MTZ. (Figure 6—figure supplement 2).

The brain activity response to the presence of paramecia could be a visual-mediated hunting response that could be mimicked by animated paramecia or other sensory responses, such as olfactory-mediated motor response. Both of which could increase the chances of successful prey capture resulting in increased food intake.

We agree that the increased “appetite” we observe could be due to an increased sensitivity to food cues that increases prey capture probability, rather than a “motivation to eat” or the actual sensation of hunger as we humans experience it. However, this concern is touching on more philosophical issues on “what it is like to be a fish” and we believe that such an explanation for the enhancement of feeding will still fall within our operational definitions of “hunger” and “satiety”. We have made this now more explicit in the text.

[Editors’ note: the author responses to the re-review follow.]

Overall, we believe the issues raised by Reviewer 2 are in some cases helpful but in others unwarranted. The criticisms mainly concern presentation and clarity and would be easily resolved by simple revisions; they are not reasonable grounds for rejection, and we respond to the various concerns below.

Reviewer #2:

I really struggled to read and understand this revised manuscript. This is particularly disappointing because many of the criticisms of the first version of the manuscript were related to a lack of clarity and details, and if anything, the revision is worse. I commend the authors for adding significant new experiments. However, most of these experiments are poorly described, appear to contain mistakes, and often cannot be evaluated. There is a general lack of rigor and quantification of key measures, including measures that were requested by reviewers, with only general statements about observations, and in some cases improper comparisons.Essential revisions:1) In the text, when the authors cite a supplemental figure, most times they do not say which panel they are referring to. This might seem like a trivial issue, but it eventually makes reviewing/reading the paper difficult. This problem is particularly acute for Figure 2—figure supplement 4 and Figure 2—figure supplement 5. These figures are quickly mentioned and not explained at all in the main text or figure legends. Other supplemental panels (e.g. Figure 2—figure supplement 1A and 1C) show data that seem not to be covered in the text. Please, in the main text, reference every panel of every figure, carefully explain the experiments and describe what they show.

We have now made clear citations to supplemental figure panels when supplemental figures are cited and made an effort throughout the main text to properly represent all of the supplemental work. However, with the editors' agreement, we do not reference nor explain every panel of every supplemental figure in the main text, as this would not be the norm. We have added to the text references to all main figure panels and made a concerted effort to increase the clarity of descriptions in our figure legends. With respect to Figure 2—figure supplements 4 and Figure 2—figure supplement 5, these figures were extensively simplified (reduced to one figure: Figure 3—figure supplement 3) and their interpretation is the subject of a paragraph in subsection “Caudal and lateral hypothalamic responses to food sensory cues are anti-correlated over short timescales” and extensive detailed elaboration in the figure legend.

Generally, we now make an explicit effort to present a cohesive narrative where every Figure and Supplemental Figure is motivated and integrated in a matter that we hope makes a lot more sense.

2) Figure 1—figure supplement 3: It is inappropriate to generate ICA data using fish from other feeding-related treatments that are not described in this manuscript. While their inclusion may be necessary to achieve statistically significant results, this is a big black box of data that cannot be evaluated by reviewers or readers. This undescribed data should either be added to the manuscript, or the ICA analysis must be removed.

We agree that the source of all image data in the Independent Component Analysis (ICA; Figure 1—figure supplement 3) should have been fully described, and that fish undergoing additional manipulations that might alter results should not be included. We now have used more stringent criteria and reduced the dataset to n = 300 fish that were either food-deprived (2 hours), or presented with food in food-deprived or fed conditions, strictly according to the experiment’s conditions. The anti-correlation between the activity of cH and LH neurons is in fact stronger with this more restricted dataset than previously. Moreover, a clear and detailed description of the analysis and data has been added to the legend of Figure 1—figure supplement 3 and also to the Materials and methods section.

3) Figure 2: The data presented in this figure suggest that 30 minutes of food deprivation is enough to cause a shift in ILH activity (when quantified by active cell count) but not in cH (when that is quantified by normalized pERK intensity). However, when ILH activity is quantified by normalized pERK intensity, 30 minutes of food deprivation is not enough to cause a shift in activity. In their response to the reviews, the authors noted that normalized pERK intensity is less precise than active cell count; perhaps if cH activity was quantifiable through cell count, 30 minutes would be enough to cause a change in that population as well. Regardless, the authors should not compare the timelines of cH and LH activity changes when using different metrics to quantify the activity of each group (as they do in –subsection “Satiation state influences the sensitivity of cH and LH populations to food”, subsection “Mutually opposing hypothalamic networks control zebrafish appetite” and Figure 2D).

In the original manuscript, we employed different metrics to quantify phospho-ERK staining intensity in the caudal (cH), medial lateral (mLH) and lateral (lLH) hypothalamus because these areas have very different distribution patterns of phospho-ERK positive cells. Labeled cells in the cH are highly clustered, and yet they are well-dispersed in the mLH and lLH. Single cell analysis, using a thresholding algorithm (Figure 1—figure supplement 4) is possible in the mLH and lLH, but cannot be automated for the cH. Such imperfections in quantification tools are unavoidable. We therefore implemented in the first revision the reviewer’s suggestion to also use the same metrics for comparisons within these areas (Figure 1G and Figure 2B). As the reviewer notes, and it is not surprising, ‘active cell count’ detects more subtle changes in LH activity than the average ROI fluorescence metric in some situations. For example, with respect to the reviewer’s specific point concerning changes in cH and lLH activity; since cH activity changes occur broadly in a dense pERK-positive population, they are readily detected as differences in average ROI fluorescence. However, since active (above threshold) neurons are sparse in the mLH and lLH of continuously fed animals, the change in average ROI fluorescence due their absence after 30 minutes of food deprivation is insignificant. The difference is, however, clearly significant in a count of active cells.

4) Subsection “Satiation state influences the sensitivity of cH and LH populations to food”: The text here is not justified by the data presented. The idea that the cH response to absence of food, which the authors claim happens after the LH response, is somehow required for LH responsiveness, does not make sense.

We agree with the reviewer and have removed this statement.

5) Figure 2—figure supplement 1A: What is the feeding condition for these graphs? What message are they intended to convey, particularly because the correlations are relatively weak (especially for cH).

Figure 2—figure supplement 1a includes data from all food deprivation times (30 minutes, 2 hours, 4 hours) that were described in Figure 2—figure supplement 1B. This is now clarified in the legend. The plot indicates that there is a weak correlation between reduced cH pERK labeling and increased food ingestion,whereas there is a more significant correlation between food ingestion and mLH/lLH activity (middle and right panels, Figure 2—figure supplement 1A). The implied message is that, since in the presence of food cH pERK activity correlates positively with satiation, food intake (and gut fluorescence) should show a negative correlation: the lower the cH activity, the more the fish eats, the higher the gut fluorescence, but this relationship may not be perfect due to biological and experimental variability.

6) Figure 2—figure supplement 2A is problematic. The top and right images are at different scales, and the two pERK images looking very different. There seem to be much fewer TH2 positive cells than in other images the authors provide (e.g. panel e of same figure). A full z-stack of the cH area should be provided.

As referred to by the reviewer, the old Figure 2—figure supplement 2a displayed single-plane images of overlap between 5-HT and pERK antibody staining and between TH2:GCaMP5 and pERK antibody staining. As was indicated by scale bars, these images were at different magnifications.

We have replaced these single examples with a more comprehensive analysis in which both pERK and 5-HT staining were performed in a TH2:GCaMP5 background (n = 4 fish). Hence, the pERK expression overlap with dopaminergic and serotonergic populations was obtained in the same images. The new data is presented in Figure 1—figure supplement 5, including a full z-series of pERK with 5-HT staining (Video 2) or with TH2:GCaMP5 (Video 3), as requested. The results support our claim that the majority of pERK-positive neurons in a food-deprived fish are serotonergic, whereas dopaminergic neurons account for a minor fraction.

7) Figure 2—figure supplement 3D: "Consistent with our pERK results, the initial calcium-mediated mean fluorescence and firing frequency of a subset of cH neurons scaled with the length of food deprivation prior to imaging (Figure 2—figure supplement 3D)". There are two problems with this statement. First, this data only shows absolute fluorescence (or is it mean fluorescence?), not firing frequency. Second, what is this "subset of cH neurons"? How many cells are quantified? Where are they? Is this a small minority of the cells, or a general feature of most cH neurons? The relevant cells must be indicated.

An error in the text created this confusion; it should have referred to Figure 2—figure supplement 4 and Figure 2—figure supplement 5 instead of Figure 2—figure supplement 3D. We also realize that the term ‘spike frequency’ should not have been used without clarification, as GCaMP fluorescence is merely correlated with voltage spikes. The ‘spikes’ depicted are spikes in GCaMP fluorescence intensity – we now refer to them as calcium/Ca^2+^ fluorescence spikes in the text. We address the issue of analysing absolute fluorescence in live imaging experiments in much more detail below.

8) Figure 2—figure supplement 4 and Figure 2—figure supplement 5: There is almost no description or explanation of the data shown in these figures, making them completely incomprehensible to this reviewer. As best as I can understand this data, there appear to be several conflicts between what is shown in different panels, with the pERK data, and what is briefly stated in the text. Maybe it's obvious for aficionados, but likely not for most readers. There are also several apparent problems. First, Figure 2—figure supplement 4 panels A and B show a blue line in the bottom line graphs – what does this correspond to? What do the different numbers indicate – # of neurons? The authors imply opposing activity patterns for lLH and cH based on Figure 2—figure supplement 4A and B (although they are not anti-correlated, just shifted relative to each other), and this relationship is not apparent in the data shown in Figure2—figure supplement 5. These figures simply cannot currently be evaluated.

We apologize for the lack of clarity in the text and the presentation of data in Figure 2—figure supplement 4 and Figure 2—figure supplement 5. We agree with the reviewer and have now revised these figures to simplify the presentation of the data and have clarified their analysis and interpretation in the text (see below our comments in the response to point 10).

To simplify the two supplemental figures they are now condensed to Figure 3—figure supplement 3. We also improved the description of all panels in the figure legends. The blue line referred to was mis-colored; it should be green (indicating mLH df/f). We regret the confusion.

Regarding whether the average calcium traces are anti-correlated rather than shifted relative to each other, as well as the consistency of this relationship, we agree that it is difficult to conclude this from the calcium-triggered averages presented. This is now indicated in the legend. However, we believe that much more convincing data with respect to the negative correlation between cH and lLH/mLH is presented in Figure 3F. We have modified the text in the Results section to be aligned with this line of reasoning.

9) Related to the last point, subsection “Satiation state influences the sensitivity of cH and LH populations to food”: "While some mLH and lLH voxels showed a predicted reduction in baseline fluorescence and firing rate, many others displayed a significant enhancement of baseline activity." It seems (Figure 2—figure supplement 4B) that most of the LH neurons (especially in the ILH) show increased activity during food deprivation using live imaging, which is the opposite of the pERK results shown in Figure 1 and Figure 2 and the outline shown in Figure 2D. Unless I am misunderstanding Figure 2—figure supplement 4 (which is quite possible), this is a major problem that is glossed over. Should the authors focus on the GCaMP results as opposed to the pERK "Given the indirect nature of activity mapping in post-fixed animals"?

We agree and have now extensively revised and simplified these two supplemental figures (Figure 2—figure supplement 4 and Figure 2 —figure supplement 5) and streamlined their explanation in the text. We removed unnecessary analysis from the figures, and aggregated data across fish, allowing them to be combined into one figure, which now appears as Figure 3—figure supplement 3.

As the reviewer noted, there is a confounding long-term increase in the baseline calcium fluorescence of the lLH region observed over the 2-hour time course of food deprivation. Baseline fluctuations in calcium reporter fluorescence can be a significant confounding factor in long-term live calcium imaging and might arise from the animal’s immobilization or the effects of irradiation with intense IR laser light. Measurements of calcium spike frequency and spike amplitude can however be made irrespective of such baseline changes, which is facilitated by baseline-subtraction (detrending). In the revised Figure 3—figure supplement 3, we have included both the raw data (Figure 3—figure supplement 3Ai) as well as baseline-detrended data for clarity (Figure 3—figure supplement 3Aii). Instead of focusing on the long-term baseline changes, we only employ live calcium imaging to look at the acute and quick changes in calcium reporter fluorescence, such as spike frequency and amplitude. Here, irrespective of the baseline change in lLH fluorescence, there is a clear difference in the activity of the cH, mLH and lLH regions in food-deprived animals as compared to live-imaged animals presented with prey (Figure 3), which we quantify and describe in the text. In these respects, our pERK and calcium imaging data are not in conflict.

With regard to the reliability of pERK-based activity measurements, these measurements are always made with reference to control samples within the same dataset, which allows for normalization to baseline activity. In addition, pERK-based activity is measured under entirely non-invasive, non-tethered and completely natural conditions in which a fixed specimen always reports activity that occurs within an approximately 15 minute window prior to sacrifice (Randlett et al., 2015).

10) Figure 4: Text describing this figure states: "exposure to this food cue in the absence of ingestion induced a small increase in lLH neural activity and a larger increase in mLH activity (Figure 4A,B). The artemia-induced hypothalamic activity was, however, less than that observed with consumable prey (Figure 4A/B)." This statement accurately describes the data. However, the next sentence: "These observations suggest that the mLH responds primarily to sensory cues and/or induced hunting behavior whereas the induction of lLH activity largely depends on consumption" is not an appropriate interpretation of the data. I would conclude that both lLH and mLH respond to both paramecia and artemia, that both populations are less responsive to artemia, and that ILH is less responsive than mLH to both stimuli, rather than that one population responds primarily to sensory cues while the other population responds to consumption. One could argue that the increase in cell counts in lLH in response to artemia is very small, however a statistically significant difference is indicated. The "active cell count" metric also seems to be flawed because the mLH area is much larger than the lLH area, and thus these values must be normalized in order to make any meaningful comparisons between the cell populations. It looks to me like normalization would likely eliminate any difference in active cell count between lLH and mLH. It is unclear if the "normalized pERK intensity" metric is similarly flawed, i.e. does this quantify the total fluorescence in the region of interest (which would be affected by the size of the region), or is this value normalized according to the area of measurement (it's unclear what "normalized" refers to here – normalized to tERK, to total area, to the control value)?

The reviewer considers that mLH and lLH activities are equivalently induced by exposure to Artemia, indicating that both areas are equivalently though modestly active in response to food sensory cues in the absence of ingestion. We have performed additional quantification to address the reviewers’ concerns, particularly whether the differences in mLH and lLH size confound our interpretation.

First, we note that since pERK fluorescence intensity is already averaged over the ROI, that ROI size has already been accounted for.

In the case of active cell count, we performed two additional analyses:

1) Quantified artemia-induced activity relative to paramecia-induced activity. Thus, the activity of each lobe is “normalized” to its maximal activity (that is, the activity induced by paramecia). Using this method, we show that indeed the lLH is still more weakly activated by food sensory cues than the mLH. This quantification is described in the text and in Figure 4.

2) Normalized “active cell count” to ROI size, as the reviewer has requested. The results are presented below. Notably, the effect of such normalization does not change the results in any significant way.

Taken together, we are now more confident in our interpretation based on the active cell count metric that the lLH is likely more responsive to consummatory cues. However, noting that the changes in overall fluorescence between the mLH and lLH are much more similar than for active cell count, we have modified our interpretation in the text to take the reviewer’s comments into account.

Finally, regarding the query about ‘normalization’, it was to the control value, hence all controls have a mean of 1. This is also the case for other pERK-related figures. We have clarified this point in the legend.

11) Figure 5E: How do the authors account for the increase in mLH GCaMP fluorescence in response to stimulation of a control area that is not labeled by ReaChR?

The reviewer is understandably puzzled by the fact 633 nm illumination of a control region (preoptic area, PO) triggers an increase in GCaMP fluorescence in the mLH (Figure 5E). We find that such illumination of the PO triggers mLH activity even in the absence of ReaChR expression. This is likely due to the fact that the preoptic area is a multisensory region that responds to visual input (Wee et al., 2019) due to the expression of many non-visual light-sensitive opsins in this region (Fernandes et al., 2012). Hence, we are not surprised that light alone may affect the activity of hypothalamic regions, in addition to the specific ReaChR induced modulation of lLH activity. All this is now explicitly discussed in the text.

12) Figure 5—figure supplement 1: As the authors note, the y333:Gal4 line is much less specific for 5-HT neurons than the 116A:Gal4 line. It is essential to determine whether the ~40% of y333:Gal4 expressing cells that are 5-HT negative are TH2 positive, as was done for the 116A:Gal4 line, since this would provide a significant dopaminergic input to this experiment.

We have now performed the requested quantification and show that 23.9 ± 2.2% (up to 30%) of y333:Gal4;UAS:ReaChR-RFP cells are dopaminergic. This data is displayed and quantified in a new Figure 5—figure spplement 2. Given that 5-HT labels ~60% of y333-labeled cells, and that we have observed cells in the cH that lack both 5-HT and DA labeling (Figure 1—figure supplement 5), the remaining cells are likely accounted for by incomplete 5-HT and DA neuron labeling, or could comprise histaminergic neurons which form a minor subset of the cH (Chen et al., 2016).

13) Figure 6—figure supplement 1: This is another example of anecdotal evidence that should be quantified.

The reviewer requests quantification of pERK staining shown in the image data of Figure 6—figure supplement 1. We now present higher quality images (n = 3 fish) and have quantified the data as requested. Our results confirm that our optogenetic behavioral setup is highly effective in activating ReaChR-positive cells.

14) Figure 6—figure supplement 2: There is no quantification of cH cell ablation. Instead, just a single exemplar image is shown. It is therefore impossible to draw any conclusions about whether or not, or to what degree, the ablation was successful. Thus, the authors' claim in the rebuttal that "We can absolutely confirm that this protocol (which we have also utilized for other transgenic lines) is sufficient to ablate most cH neurons (see Figure 6—figure supplement 2)" is in no way substantiated by the data provided.

We agree that it is best to quantify the ablation and have done so. Cells that express the UAS-nfsb:mCherry construct under control of 116A-Gal4 are visible as mCherry-positive cells. When such fish were incubated with the drug MTZ, which causes ablation, 6.1+/- 0.66 cH cells per fish were mCherry-positive (n = 54 fish). When MTZ was omitted, 31 +/- 1.5 cells were mCherry-positive (n = 3 fish). The reduction resulting from ablation was thus ~80%. The remaining mCherry-positive cells were generally dim and misshapen/deformed, indicating damage that might impair function. This information is now included in the Materials and methods section.

15) The authors propose that "optogenetic stimulation of cH activity inhibits lLH activity and thereby causes the feeding rate to decrease." (subsection “Functional dissection of cH serotonergic neurons in feeding behaviour”). They provide evidence that cH activation inhibits feeding (although that needs to be clarified; see comment below), and that cH activation also inhibits ILH activity. Is there any causal evidence that reduced ILH activity reduces feeding? If not, this statement should be deleted.

The reviewer queries whether there is evidence that lLH activity is required for feeding behavior. This was in fact examined by Muto et al., 2017 who ablated LH neurons and observed reduced feeding behavior (Figure 1F, Muto et al., 2017). This was previously mentioned in the Discussion section, but we have now reiterated it right below the referenced statement.

16) Despite the caveats mentioned by the authors, the use of different Gal4 lines with different expression patterns (which remain lightly characterized despite reviewer requests for more details) for genetic ablation and optogenetic activation precludes the authors from drawing firm conclusions from these experiments. It doesn't really make sense that a Gal4 line would be strong enough to drive cell ablation (especially since the authors use an unusually low concentration of mtz) but not optogenetic stimulation. Just because a Gal4 line doesn't produce the hoped for phenotype (for which the data is not shown) does not mean that one can simply substitute another Gal4 line that does produce the hoped for result.

The reviewer offers the opinion that the Gal4 lines used in this study were “lightly characterized”. Importantly, contrary to the reviewer’s stated concern, the reason the 116A:Gal4 line could not be used in optogenetic experiments is that it failed to drive expression of the UAS:ReaChR-RFP transgene as well as many other existing UAS-driven variants of channelrhodopsin (UAS:ChR2-YFP, UAS:ChRWR-GFP, UAS:ChR2:mCherry). This was noted by virtue of failing to observe the fluorescence tag in 116A:Gal4; UAS-Channelrhodpsin animals, even after post-hoc staining for signal amplification was performed. As a result, optogenetic experiments could not be attempted with the 116A:Gal4 line. We also found that UAS:nfsb-mCherry expression is relatively weak compared to UAS:GFP expression driven by 116A:Gal4. Overall, the reviewer’s assertion that we substituted another Gal4 line because we did not observe the “hoped for phenotype” is unwarranted. Our decisions regarding the choice of Gal4 lines were based entirely on tests of transgene expression.